# Principles of paralog-specific targeted protein degradation engaging the C-degron E3 KLHDC2

Daniel C. Scott[1,6], Suresh Dharuman[2,6], Elizabeth Griffith[2], Sergio C. Chai[2], Jarrid Ronnebaum[2], Moeko T. King[1], Rajendra Tangallapally[2], Chan Lee[3], Clifford T. Gee[2], Lei Yang[2], Yong Li[2], Victoria C. Loudon[2], Ha Won Lee[2], Jason Ochoada[2], Darcie J. Miller[1], Thilina Jayasinghe[2], Joao A. Paulo[3], Stephen J. Elledge[4], J. Wade Harper[3], Taosheng Chen[2], Richard E. Lee[2] ✉ & Brenda A. Schulman[1,5] ✉

PROTAC® (proteolysis-targeting chimera) molecules induce proximity between an E3 ligase and protein-of-interest (POI) to target the POI for ubiquitin-mediated degradation. Cooperative E3-PROTAC-POI complexes have potential to achieve neo-substrate selectivity beyond that established by POI binding to the ligand alone. Here, we extend the collection of ubiquitin ligases employable for cooperative ternary complex formation to include the C-degron E3 KLHDC2. Ligands were identified that engage the C-degron binding site in KLHDC2, subjected to structure-based improvement, and linked to JQ1 for BET-family neo-substrate recruitment. Consideration of the exit vector emanating from the ligand engaged in KLHDC2's U-shaped degron-binding pocket enabled generation of SJ46421, which drives formation of a remarkably cooperative, paralog-selective ternary complex with BRD3[BD2]. Meanwhile, screening pro-drug variants enabled surmounting cell permeability limitations imposed by acidic moieties resembling the KLHDC2-binding C-degron. Selectivity for BRD3 compared to other BET-family members is further manifested in ubiquitylation in vitro, and prodrug version SJ46420-mediated degradation in cells. Selectivity is also achieved for the ubiquitin ligase, overcoming E3 auto-inhibition to engage KLHDC2, but not the related KLHDC1, KLHDC3, or KLHDC10 E3s. In sum, our study establishes neo-substrate-specific targeted protein degradation via KLHDC2, and provides a framework for developing selective PROTAC protein degraders employing C-degron E3 ligases.

Targeted protein degradation (TPD) is an important emerging strategy in drug discovery and chemical biology[1]. The PROTAC (Proteolysis Targeting Chimera) class of molecules achieve degradation by inducing proximity between a protein-of-interest (POI) and an E3 ligase[2–4]. PROTAC protein degraders consist of ligands individually binding each entity, connected by a chemical linker[5]. A productive POI-PROTAC-E3 ternary complex renders the POI a neo-substrate for the E3 ligase. After ubiquitin-mediated proteolysis of the neo-substrate, the PROTAC becomes available to target another POI molecule to the E3. As they are turned over multiple times, PROTAC protein degraders possess

catalytic-like activity in cells prompting failure to the targeted biological system, which can only be reversed by resynthesis of the POI[6,7].

Despite the existence of ~ 600 E3 ligases in humans, only a handful have been successfully co-opted by PROTAC protein degraders[8–30]. The majority of these belong to the largest family of E3s, the Cullin-RING ligases (CRLs)[31]. CRLs are attractive handles for degrader development owing to their modular mix-and-match nature[31,32]. Variable "substrate receptor" modules with distinct substrate-binding surfaces bind interchangeably at one end of elongated cullin proteins (Fbox protein-SKP1 complexes with CUL1, a subset of BCbox-ELOB-ELOC complexes with CUL2, BTB proteins with CUL3, DCAF-DDB1 complexes with CUL4, and another subset of BCbox-ELOB-ELOC complexes with CUL5)[32–38]. The opposite end of the cullin binds a dedicated E3 ligase RING domain subunit (RBX1 for CUL1-CUL4, and RBX2 for CUL5). When the cullin is neddylated, the cullin-RING complex partners with a ubiquitin carrying enzyme that ubiquitylates the receptor-bound substrate[39,40]. The combination of hundreds of substrate receptors, five major CUL-RBX complexes, and nearly 10 different ubiquitin-carrying enzymes offers a plethora of catalytic geometries for PROTAC-mediated ubiquitylation[41,42]. As such, there is great interest in generating ligands harnessing different E3 ligases with varying cellular expression patterns so as to provide diversity for POI engagement[43–47].

Most of our understanding of PROTAC-mediated TPD has come from studies employing VHL and CRBN. These substrate receptors assemble with CUL2-RBX1 and CUL4-RBX1, respectively, to form CRL2$^{VHL}$ and CRL4$^{CRBN}$ E3s. PROTAC degrader molecules are often thought to flexibly tether independently-engaged E3 and POI. However, formation of such an E3-PROTAC-POI ternary complex is often insufficient to confer selective ubiquitin-mediated degradation. Classic studies of E3 ligases demonstrated that ubiquitylation depends on substrate positioning on the E3 with surface lysine(s) accessible to the ubiquitylation active site[39,48–51]. Furthermore, while CRBN-POI ternary complexes do not necessarily need to possess cooperative interactions for efficient target degradation[52,53], some VHL-POI ternary complexes are thought to achieve high target selectivity and degradation due to ternary complexes exhibiting positive cooperativity[54–57].

CRL2$^{KLHDC2}$ has been proposed as an attractive additional E3 ligase handle for targeted protein degradation[43]. KLHDC2 is amongst the suite of E3s shown to bind peptide-like sequences at protein termini[58–65]. Specifically, KLHDC2 recognizes substrates possessing a Gly-Gly C-terminal degron (C-degron) sequence[61,62]. Several properties of KLHDC2 suggest amenability for PROTAC development. First, crystal structures showed KLHDC2 binds substrate C-termini in a well-defined pocket lined by aromatic residues available to engage a ligand[66]. Second, the C-degron consensus sequence binding CRL2$^{KLHDC2}$ was discovered based on its conferring ubiquitin-dependent degradation when C-terminally fused to GFP[61,62,67,68]. Such an arrangement might be successfully mimicked by a PROTAC simply tethering a POI. Indeed, proteome wide screens surveying tens or more E3 ligases for the potential to confer degradation on GFP or GFP-tagged neo-substrates identified CRL2$^{KLHDC2}$ as a top hit[69,70]. Moreover, a fusion of a peptide derived from a KLHDC2 substrate to a pan kinase inhibitor led to the degradation of numerous kinases[71]. PROTACs reported by Arvinas[72], and patent WO 2023/192578 A1 reported by Kymera, during the preparation of our manuscript further validated the possibility of co-opting KLHDC2 for TPD.

Despite the many favorable properties of KLHDC2, this system also presents hurdles as well as features whose impact on degrader development is unknown. Most importantly, in natural substrates, the C-terminal carboxylate is a major determinant of binding selectivity and affinity[61,62,66]. Therefore, small molecules targeting KLHDC2's substrate binding pocket may favor acidic substituents, which are known to hinder cell permeability. Also, several closely related KLHDC-family E3s recognize a substrate's C-terminal Gly, raising questions as to whether PROTAC protein degraders can be generated that are truly

selective for KLHDC2[61,62,73–77]. In addition, KLHDC2 is subjected to additional layers of regulation beyond the typical neddylation/deneddylation pathway[78,79]. Substrate binding is subject to a kinetic filter: the substrate must bind with a fast enough on-rate to capture a KLHDC2-ELOB-ELOC 'monomer' in equilibrium with an autoinhibited tetramer (i.e. a complex of 4 KLHDC2-ELOB-ELOC protomers)[79]. Substrate binding drives E3 ligase activation in a feed-forward manner[79]. Although this regulatory mechanism amplifies KLHDC2 selectivity for bonafide substrates versus non-substrates with C-degron-like sequences, such inherent auto-inhibition of an E3 ligase is not a known feature of other CRLs employed by a PROTAC. Finally, unlike VHL and CRBN, which bind degrader molecules in relatively shallow grooves within a dozen angstroms from the protein surface, KLHDC2 engages a substrate's C-terminus within a deep groove more than 20 Å from the surface[13,54,66,80]. This architecture limits potential substrate trajectories and could impact achieving selective and/or cooperative CRL2$^{KLHDC2}$-PROTAC-POI interactions. This challenge is highlighted by the non-cooperative PROTAC-mediated interactions between CRL2$^{KLHDC2}$ and various BRD proteins reported during the preparation of our manuscript[72].

Here, we show the feasibility of selectively employing KLHDC2 for target degradation starting with ligand discovery based on small molecule screening. Utilizing the BRD targeting ligand JQ1 as a proof-of-concept bait and structure-based PROTAC elaboration, we gained insights into rules for cooperative and selective POI engagement by KLHDC2. We report methods to attain a cooperative KLHDC2-PROTAC-POI ternary complex with DC$_{50}$ and D$_{max}$ values rivaling those previously reported for JQ1-based PROTAC protein degraders co-opting VHL or CRBN. Moreover, our PROTAC specifically engaging KLHDC2 achieves remarkable POI selectivity, favoring BRD3 over BRD2 and BRD4, both for cooperative ternary complex formation and degradation upon PROTAC titration in cells.

## Results

### Discovery of a small molecule ligand targeting KLHDC2's di-Gly binding pocket

In an attempt to identify non-peptide ligands that target KLHDC2's substrate binding domain (SBD), we developed a thermal shift assay probing for stabilization of KLHDC2$^{SBD}$ upon ligand addition. The KLHDC2$^{SBD}$ alone displayed a melting temperature of 40.9 °C. Addition of a peptide derived from the substrate SELK stabilized the KLHDC2$^{SBD}$ and increased the melting temperature to 55.0 °C (Fig. 1a). We utilized the thermal shift assay to screen ~ 9,000 molecules from a diverse collection of lead-like molecules (Figs. 1b, S1a, Supplementary Table 1). Interestingly, a portion of the initial hit compound SJ6145, which elicited an 1.1 °C thermal shift, matches a C-terminal di-Gly (Figs. 1c, S1b). SJ6145 inhibited KLHDC2$^{SBD}$ binding to a di-Gly degron peptide, with an IC$_{50}$ value of 42 μM as measured by a TR-FRET assay (Figs. 1c, S1c). Surface Plasmon Resonance (SPR) confirmed that SJ6145 binds directly to KLHDC2$^{SBD}$ (K$_d$ = 16.9 μM, Figs. 1c, S1d, Supplementary Table 2).

### Hit to lead optimization of SJ6145

The initial hit compound, SJ6145, was subjected to extensive structure-activity relationship (SAR) analyses (Fig. 1d, Supplementary Table 2). As a di-Gly motif is known to significantly influence binding to KLHDC2, this moiety was retained throughout the SAR investigation. Substituting the central thiazole ring with triazole and imidazole rings resulted in a reduction in potency compared to the original hit SJ6145 (Fig. 1d, Supplementary Table 2). Subsequently, efforts were directed towards optimizing the exterior phenyl ring portion to enhance potency. Notably, various substituents (meta-Cl, -F, -Me, for affinity, and para-methoxymethyl groups as suitable exit vector for PROTAC development) on the phenyl ring were well tolerated (Fig. 1d, Supplementary Table 2). However, the introduction of a nitrogen atom or

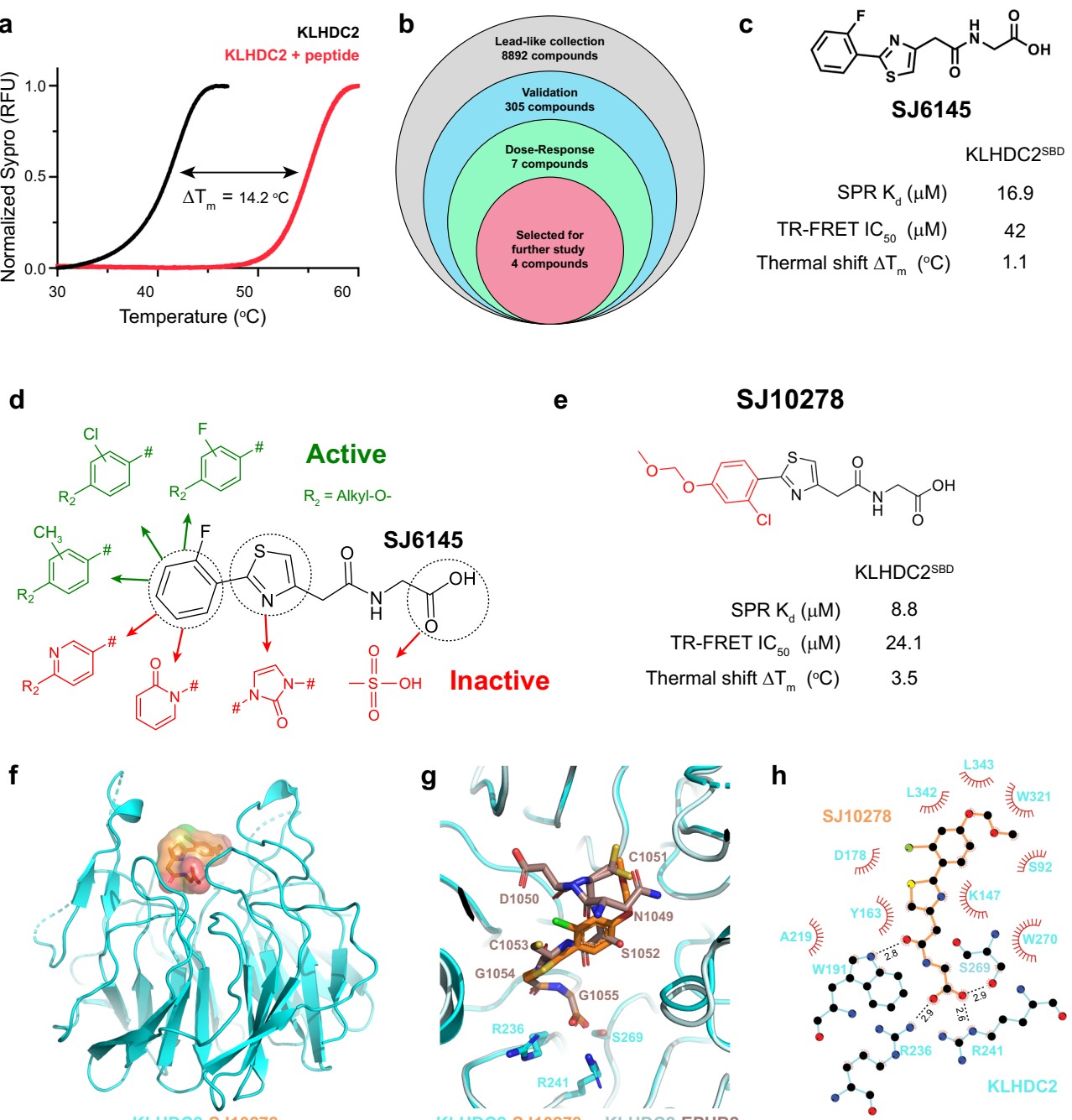

**Fig. 1 | Discovery of a ligand targeting KLHDC2. a** Thermal shift assay monitoring the denaturation of the KLHDC2$^{SBD}$ in the absence (black) or presence (red) of a diGly substrate peptide from SELK. The change in melting temperature $\Delta T_m$ is shown. **b** Summary of workflow for the thermal shift based small molecule screening campaign. **c** Chemical structure and biophysical characterization of hit SJ6145. Data are the average of $n = 2$ independent experiments. **d** Summary of SAR campaign surrounding SJ6145. **e** Chemical structure and biophysical characterization of lead molecule SJ10278. Data are the average of $n = 2$ independent

experiments. **f** Crystal structure of KLHDC2$^{SBD}$ (cyan, cartoon) bound to SJ10278 (orange, surface). **g** Structural superposition of SJ10278-KLHDC2$^{SBD}$ (SJ10278 in orange and KLHDC2$^{SBD}$ in cyan) with the structure of KLHDC2$^{SBD}$ bound to a diGly peptide from EBHB2 (KLHDC2$^{SBD}$ in pale cyan and EBHB2 peptide in brown; 8EBL.pdb). The C-terminal RSR recognition motif from the KLHDC2$^{SBD}$ is shown in sticks. **h** LigPlot of SJ10278-KLHDC2$^{SBD}$, hydrogen bonds between SJ10278 and KLHDC2$^{SBD}$ are shown in dashed black lines with distances indicated. Source data are provided as a Source Data file.

disruption of aromaticity on the ring led to a loss of activity (Supplementary Table 2). This SAR campaign led to our first lead, SJ10278, with enhanced potency compared to SJ6145 as measured by SPR ($K_d = 8.8 \,\mu M$), competitive binding to a degron peptide (TR-FRET IC$_{50}$ = 24.1 $\mu M$), and thermal shift ($\Delta T_m = 3.5\,°C$; Figs. 1e, S1e-g, Supplementary Table 2).

To understand how SJ10278 engages KLHDC2's di-Gly binding pocket, we obtained a crystal structure at 1.95 Å resolution (Figs. 1f, S1h, i,

Supplementary Table 3). Superposition with the structure of a di-Gly degron peptide confirmed the ligand targets KLHDC2's substrate-binding pocket. The acid moiety of SJ10278 is stabilized by canonical contacts with KLHDC2's R236, S269, and R241 (Fig. 1g). The thiazole and chlorophenyl moieties of SJ10278 contact adjacent hydrophobic and polar elements typically engaged by consensus degrons (Fig. 1g, h). However, attempts to substitute the SJ10278 carboxylate with a sulfonate isostere led to a loss of activity.

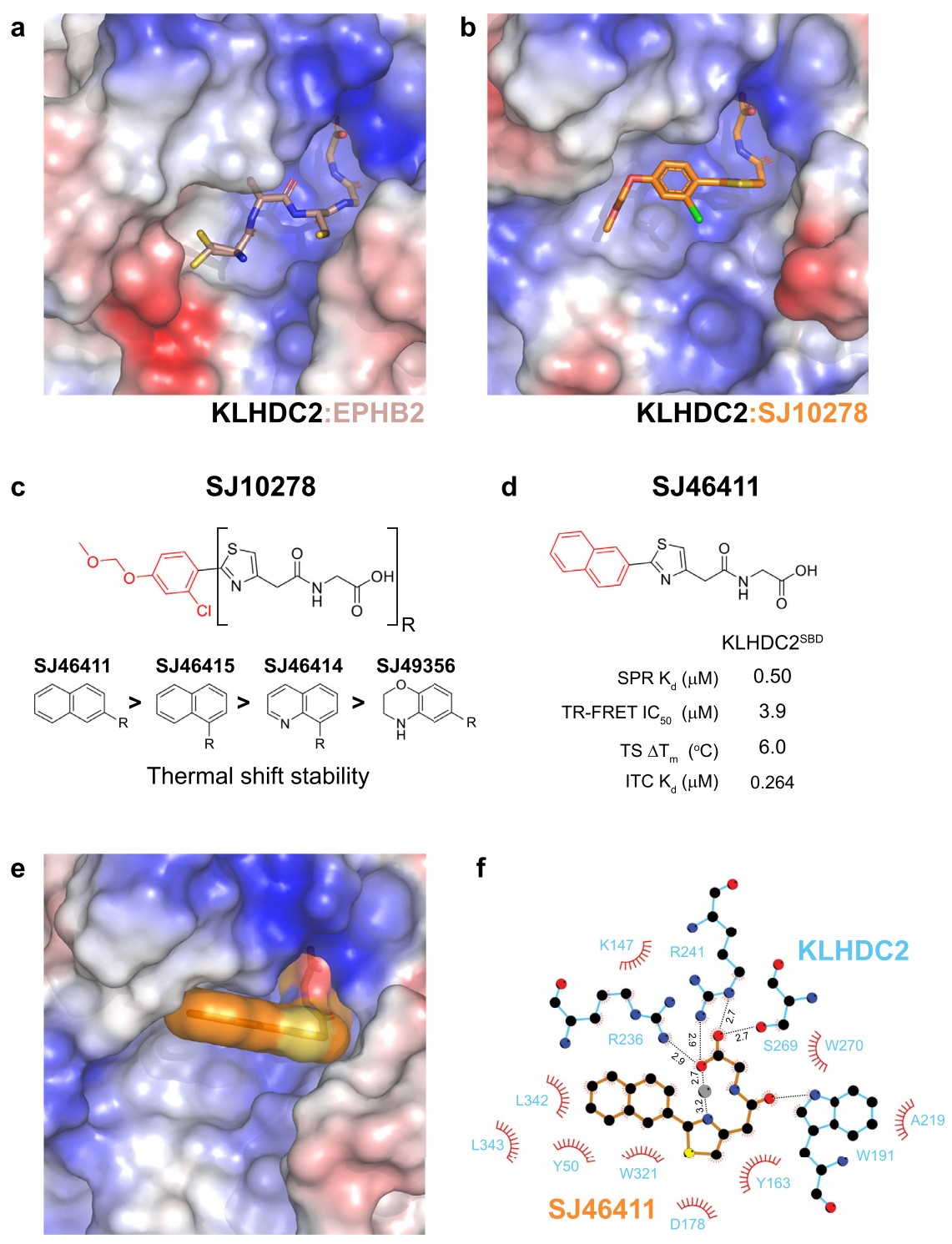

**Fig. 2 | Optimization of a KLHDC2 targeting ligand. a** Electrostatic representation of KLHDC2SBD bound to the diGly substrate EPHB2 (chocolate, sticks, 8EBL.pdb). For clarity, only the C-terminal five residues of EPHB2 are shown. **b** As in (**a**), but with SJ10278 (orange, sticks). **c** Summary of SAR campaign surrounding SJ10278. **d** Chemical structure and biophysical characterization of SJ46411. Data are the average of $n = 2$ independent experiments. **e** As in (**a**) but with SJ46411 shown in surface representation. **f** LigPlot of SJ46411-KLHDC2SBD, hydrogen bonds between SJ46411 and KLHDC2 are show in black sticks with distances indicated. The water molecule involved in intramolecular hydrogen bonding with SJ46411 is shown as a gray sphere. Source data are provided as a Source Data file.

Two features of the SJ10278-KLHDC2 co-crystal structure were notable for structure guided improvement of ligand potency. First, the constellation of hydrophobic residues surrounding the chlorophenyl moiety suggested adding hydrophobic character to this portion of the ligand might improve binding (Fig. 1h). This is particularly evident in electrostatic representations of the binding pocket surface, which shows SJ10278 adjacent to a hydrophobic cleft in KLHDC2 (Fig. 2a, b). Secondly, comparison to a bound consensus di-Gly substrate showed that binding of SJ10278 leaves space in the pocket (Fig. 2a, b). We therefore tested if expanding the phenyl group

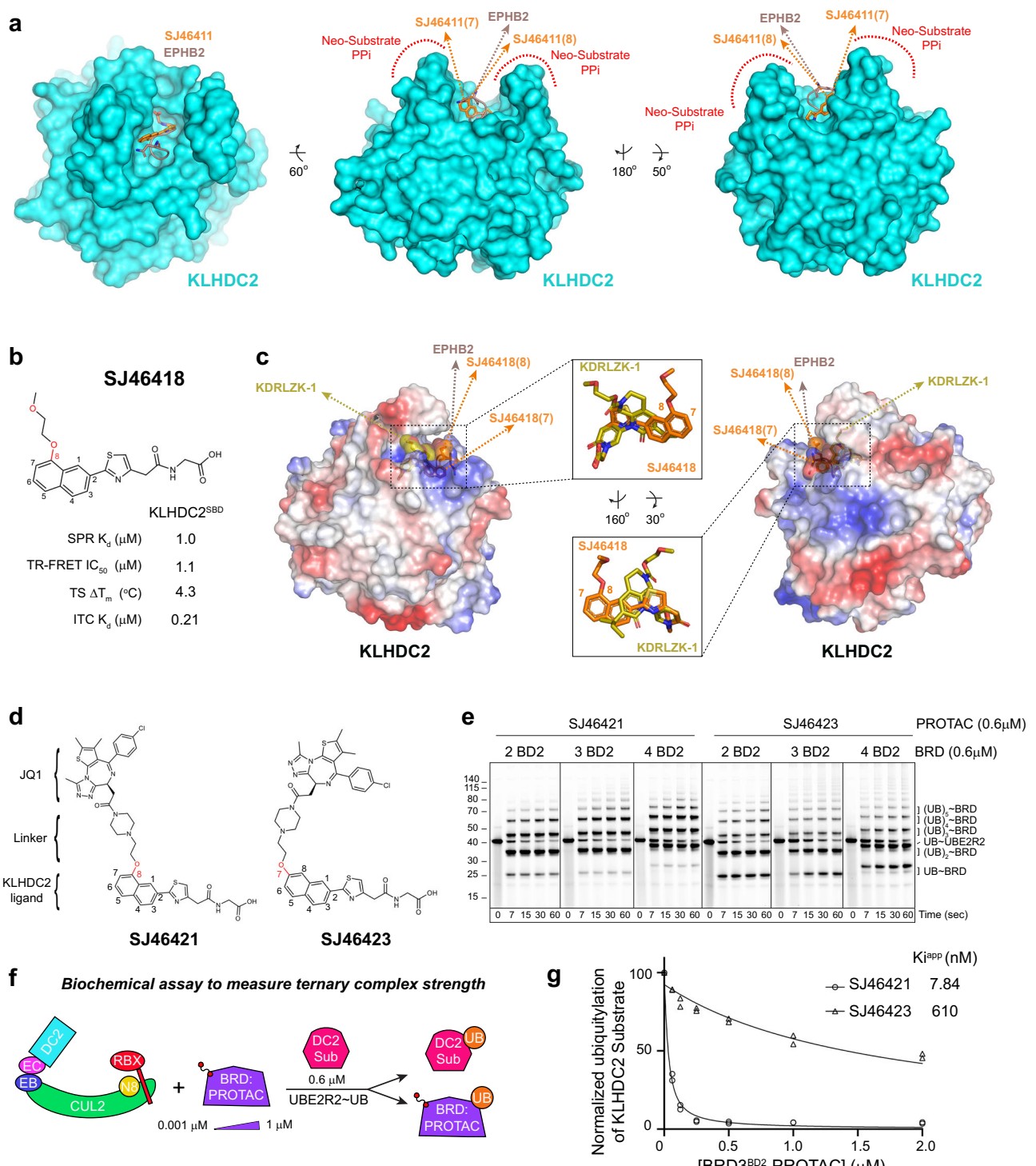

into naphthyl or bi-aryl analogs, which could potentially better fill this pocket, would improve binding (Fig. 2c, Supplementary Tables 4 and 5). Amongst the various naphthyl analogs tested, SJ46411 exhibited excellent activity with a $K_d$ of ~260 nM measured by isothermal titration calorimetry (SPR $K_d$ = 0.5 µM, TR-FRET $IC_{50}$ = 3.9 µM, $\Delta T_m$ = 6.0 °C, Figs. 2c, d, S2a–f). We also note that the 2-naphthyl substitution demonstrated greater activity than the 1-naphthyl substitution (SJ46411 vs SJ46145, Supplementary Table 4). Also, the morpholino analog SJ49356 showed less potency compared to other analogs, suggesting that molecular planarity and a lack of hydrogen bonding acceptors may be advantageous for binding (Fig. 2c, Supplementary Table 4).

A co-crystal structure of SJ46411 bound to KLHDC2 confirmed the predicted stacking of the hydrophobic face of the naphthyl ring in the binding pocket (Figs. 2e, f, S2g, h, Supplementary Table 3). The structure further showed the di-Gly portion of the molecule allows a water-mediated bridge between the thiazole ring and acid moieties (Fig. 2f).

**Generation of PROTAC protein degraders employing KLHDC2**
Having successfully generated a ligand with sufficient binding affinity, we sought to (1) devise a degrader employing KLHDC2, and (2) in so doing systematically examine rules of engagement. We considered the unique architecture of the KLHDC2 substrate-binding cleft, which surrounds one end of the ligand (the di-Gly motif), and encapsulates it

**Fig. 3 | Generation of a PROTAC employing KLHDC2. a** Structural superposition of KLHDC2$^{SBD}$ (cyan, surface) bound to SJ46418 (orange, sticks) with the diGly substrate EPHB2 (chocolate, sticks, 8EBL.pdb). For clarity surface representation is depicted for the KLHDC2$^{SBD}$ bound to SJ46418 only. Probable exit vectors from the ligands are depicted by arrows. Potential neo-substrate interaction surfaces with the KLHDC2$^{SBD}$ U-shaped binding cleft is shown. **b** Chemical structure and biophysical characterization of SJ46418. Data are the average of $n = 2$ independent experiments. **c** Structural superposition of the KLHDC2$^{SBD}$ (electrostatic, surface) bound to SJ46418 (orange, surface) to the diGly substrate EPHB2 (chocolate, surface, 8EBL.pdb) and the KLHDC2 targeting ligand KDRLZK1-KLHDC2$^{SBD}$ (yellow, surface, 8SGE.pdb). For clarity surface electrostatic representation is depicted for KLHDC2$^{SBD}$ bound to SJ46418 only. Probable exit vectors deriving from attachment sites to the ligands is depicted by arrows. **d** Chemical structure of JQ1 based KLHDC2 PROTAC protein degraders SJ46421 and SJ46423. **e** Fluorescent scan of gel

monitoring SJ46421- and SJ46423-dependent ubiquitylation of the BD2 domain from BRD2, BRD3, or BRD4. Reactions were performed in pulse-chase format, with fluorescent ubiquitin charged-UBE2R2 added to neddylated CRL2$^{KLHDC2}$ and indicated PROTAC/BRD mixture. Shown is representative panels from $n = 2$ independent experiments. **f** Cartoon depiction of the biochemical assay monitoring inhibition of KLHDC2 di-Gly substrate ubiquitylation by PROTAC-mediated ternary complex formation. Briefly, neddylated CRL2$^{KLHDC2}$ was incubated with BRD3$^{BD2}$ and the indicated PROTAC prior to adding fluorescent ubiquitin-charged UBE2R2 (i.e. the pre-formed thioester-linked UBE2R2-ubiquitin intermediate) and a di-Gly protein substrate. Loss of di-Gly protein substrate ubiquitylation was quantified. **g** Morrison fit of data from competition ubiquitylation assay, yielding a Ki$^{app}$ as a measure of the strength of ternary complexes promoted by SJ46421-BRD3$^{BD2}$ and SJ46423-BRD3$^{BD2}$. Data are the average from $n = 2$ independent experiments. Source data are provided as a Source Data file.

at the bottom of a deep U-shaped cleft. We hypothesized that the pillar-like sides of the cleft could create or limit opportunities for PROTAC-mediated E3-POI complex formation (Fig. 3a). Thus, the resultant exit vector from the KLHDC2-binding ligand could substantially impact cooperativity and PROTAC effectiveness and selectivity for POI engagement.

To evaluate potential exit vectors, we compared structures of KLHDC2 bound to SJ46411 and to a di-Gly substrate peptide (Fig. 3a)[79]. This analysis revealed that linker attachment to position 8 of the naphthyl ring would generate an exit vector closely matching that of a natural substrate. Meanwhile, substitution at position 7 seemed less ideal for placing a POI in a location corresponding to where CRLs typically bind substrates. To gain structural insights into a potentially suitable ligand exit vector, we synthesized and characterized SJ46418, a derivative of SJ46411 with an alkyl linker at position 8 of the naphthyl ring (Fig. 3a, b, Supplementary Table 4). Biochemical and structural analysis of SJ46418 demonstrated that the introduced linker did not perturb affinity, nor alter the binding mode towards KLHDC2 (SPR $K_d = 1.0\,\mu M$, TR-FRET IC$_{50} = 1.1\,\mu M$, $\Delta T_m = 4.3\,°C$, Figs. 3b, S3a–h). We also performed retrospective structural comparison to a KLHDC2 ligand (KDRLZK-1) reported during manuscript preparation[72]. Of note, the resulting exit vectors from substituents at either position 7 or 8 of the naphthyl ring in our ligand differ substantially from that provided by KDRLZK-1 (Fig. 3c).

Using the bromodomain targeting ligand JQ1 as a test case, we linked it to positions 7 or 8 of the naphthyl ring via a ethoxypiperazine linker to generate SJ46421 and SJ46423 (Fig. 3d, as described in Supplementary Methods). As a first test of activity of these molecules, we asked if they could recruit and promote ubiquitylation of BRD-family neo-substrates. These initial ubiquitylation assays used a modified version of neddylated CRL2$^{KLHDC2}$ that is not subject to autoinhibition[79]. Both SJ46421 and SJ46423 were able to promote polyubiquitylation of the BD2 domain from BRD2, BRD3, or BRD4 (Fig. 3e). Notably, the ubiquitin chains were relatively longer in reactions employing SJ46421 (Fig. 3e). Greater numbers of ubiquitin modifications typically reflect longer residence time of a POI on an E3.

How does ubiquitylation of a PROTAC-mediated neo-substrate compare to that of a KLHDC2 substrate? And how much more effective is SJ46421 than SJ46423? We addressed these questions by performing a competition ubiquitylation assay, testing effects of adding a PROTAC and BRD3$^{BD2}$ neo-substrate on ubiquitylation of a KLHDC2 di-Gly substrate. Briefly, increasing concentrations of 1:1 mixtures of SJ46421:BRD3$^{BD2}$ or SJ46423:BRD3$^{BD2}$ were incubated with neddylated CRL2$^{KLHDC2}$ E3. After 15 minutes, ubiquitylation reactions were initiated by simultaneously adding the KLHDC2 substrate and the ubiquitin-loaded E2 (i.e., thioester-linked UBE2R2-ubiquitin). If the PROTAC:BRD3$^{BD2}$ occupies KLHDC2's SBD, then substrate ubiquitylation would be inhibited (Fig. 3f). Utilizing this approach, it was apparent that SJ46421 was able to promote a substantially tighter

ternary complex with KLHDC2, as its inhibition was two-orders of magnitude greater than that of SJ46423 (IC$_{50}$ 7.8 nM and 610 nM for SJ46421 and SJ46423 respectively, Figs. 3g, S3i).

## SJ46421 promotes cooperative paralog selective ternary complex formation

A wealth of data indicates that the most effective VHL-based PROTAC protein degraders promote positively cooperative ternary complexes between the E3 ligase and the POI target[54,81–84]. The cooperative effects on binding affinity result from neo interactions between the POI and E3, and/or between either or both proteins and the ligand beyond their individual binding moieties[54,85]. Such interactions enhance affinity for the target POI, and can establish paralog selectivity[53,86–91]. Interrogation of cooperative PROTAC-mediated ternary complex was pioneered in studies of the PROTAC MZ1, which connects JQ1 to a ligand employing VHL. Cooperativity of complex formation is defined by the parameter alpha ($\alpha = (K_d(Binary)/K_d(Ternary))$), with positively cooperative interactions having values larger than one, and non-cooperative or uncooperative interactions having values equal to one or less than one, respectively[54].

Strikingly, ternary complex formation only between SJ46421 and BRD3$^{BD2}$ is highly cooperative ($\alpha = 16$; Figs. 4a, S4, Supplementary Table 6). Ternary complexes between SJ46421 and BRD4$^{BD2}$ or BRD2$^{BD2}$ were moderately cooperative ($\alpha = 2.8$ and 1.4 respectively; Figs. 4a, S4, Supplementary Table 6). Notably, the most cooperative interaction also promoted the most stable ternary complex [$\Delta G$ (binary + ternary) of −19.9 kcal/mol for BRD3$^{BD2}$ versus −18.4 kcal/mol for BRD2$^{BD2}$].

Given the strong cooperative nature of SJ46421, we wondered if the comparatively poor ternary complex affinity with SJ46423 might be due to a reduced ability to promote cooperative interactions. To test this, we first measured affinities for SJ46423 alone and BRD3$^{BD2}$-SJ46423 to KLHDC2 via ITC. The results indicated that SJ46423 did indeed promote a less positive cooperative ternary complex between BRD3$^{BD2}$ and KLHDC2 ($\alpha = 2.9$; Figs. 4a, S4, Supplementary Table 6). However, the N values associated with these titrations were low (~0.5), presumably due to the poor solubility of SJ46423 at 40 µM in the sample cell. We therefore attempted to validate these observations through additional biochemical studies. To do so, we compared the inhibition of KLHDC2 substrate ubiquitylation by ligands alone, or in comparison to their respective ternary complexes with BRD3$^{BD2}$. Strong positive enhancement of inhibition was readily observed when comparing SJ46421 to SJ46421-BRD3$^{BD2}$, with ~2 orders-of-magnitude improvement in the IC$_{50}$ (Fig. 4b, c). In contrast, SJ46423-BRD3$^{BD2}$ was only able to enhance the IC$_{50}$ ~7 fold in comparison to SJ46423 alone (Fig. 4b, c). Taken together, our data suggest that considering the directionality of the ligand's exit vector from KLHDC2's binding pocket provides an opportunity to generate a PROTAC mediating cooperative interactions between this E3 and neo-substrates.

**a**

| Syringe | Cell | $K_d$ (nM) | $\Delta G$ | $\Delta H$ | N | $\alpha$ |
|---|---|---|---|---|---|---|
| BRD2[BD2] | SJ46421 | 2090 ± 279 | -7.93 | -10.3 ± 0.428 | 0.7 | - |
| BRD3[BD2] | SJ46421 | 987 ± 111 | -8.39 | -13.5 ± 0.307 | 0.8 | - |
| BRD4[BD2] | SJ46421 | 1020 ± 88.2 | -8.36 | -9.94 ± 0.178 | 0.8 | - |
| KLHDC2[SBD] | SJ46421 | 304 ± 49 | -9.10 | -8.07 ± 0.219 | 1.0 | - |
| KLHDC2[SBD] | BRD2[BD2]-SJ46421 | 208 ± 15 | -9.33 | -10.4 ± 0.100 | 0.9 | 1.4 |
| KLHDC2[SBD] | BRD3[BD2]-SJ46421 | 19 ± 3 | -10.8 | -6.65 ± 0.070 | 0.9 | 16 |
| KLHDC2[SBD] | BRD4[BD2]-SJ46421 | 109 ± 11 | -9.72 | -8.22 ± 0.090 | 0.8 | 2.8 |
| KLHDC2[SBD] | SJ46423 | 740 ± 70 | -8.61 | -3.35 ± 0.109 | 0.4 | - |
| KLHDC2[SBD] | BRD3[BD2]-SJ46423 | 251 ± 60 | -9.33 | -4.41 ± 0.075 | 0.4 | 2.9 |

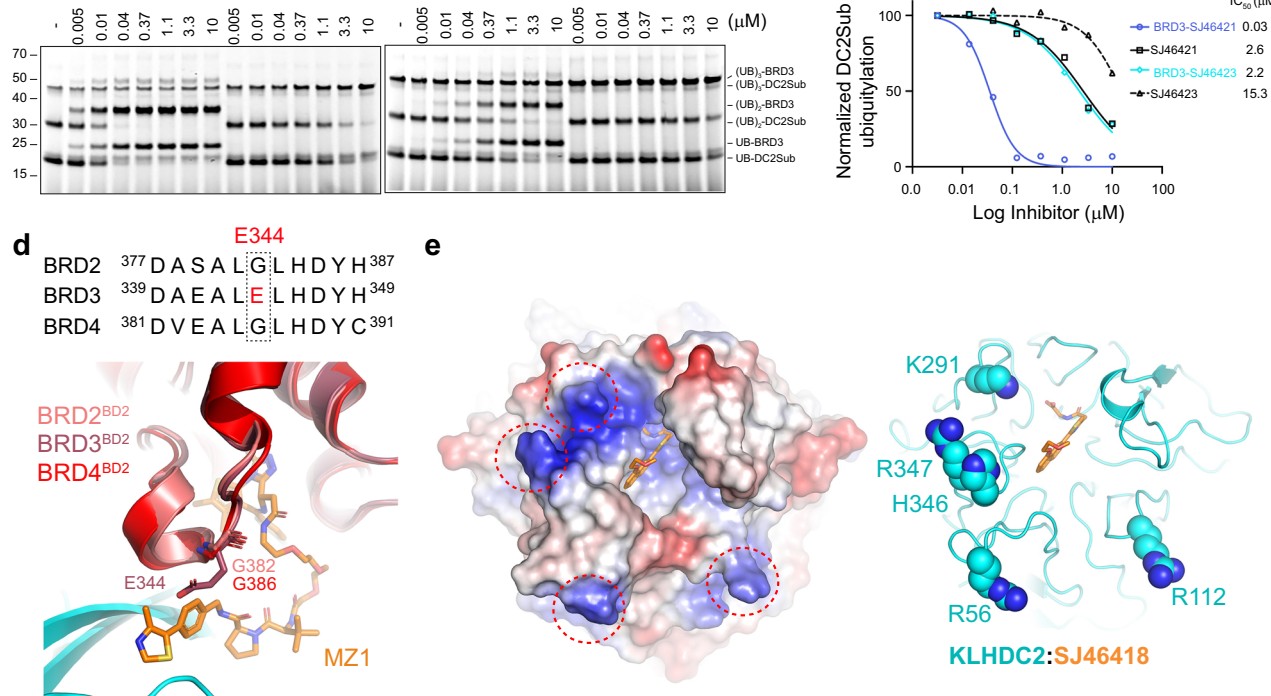

**f**

| Syringe | Cell | $K_d$ (nM) | $\Delta G$ | $\Delta H$ | N | $\alpha$ |
|---|---|---|---|---|---|---|
| KLHDC2[SBD] | G382E BRD2[BD2]-SJ46421 | 95.4 ± 10.0 | -9.80 | -10.5 ± 0.110 | 0.9 | 3.2 |
| KLHDC2[SBD] | E344G BRD3[BD2]-SJ46421 | 181 ± 8.8 | -9.42 | -7.19 ± 0.043 | 1.0 | 1.7 |
| KLHDC2[SBD] | G386E BRD4[BD2]-SJ46421 | 24.5 ± 1.7 | -10.6 | -9.05 ± 0.037 | 0.9 | 13 |
| R56A KLHDC2[SBD] | SJ46421 | 463 ± 15 | -8.85 | -1.82 ± 0.051 | 1.0 | - |
| R56A KLHDC2[SBD] | BRD3[BD2]-SJ46421 | 273 ± 25 | -9.19 | -3.68 ± 0.053 | 1.1 | 1.7 |
| R56A KLHDC2[SBD] | G386E BRD4[BD2]-SJ46421 | 329 ± 5.6 | -9.05 | -4.69 ± 0.15 | 1.1 | 1.4 |

What drives the selectivity toward BRD3[BD2]? In sequence alignments between BRDs, E344 in BRD3[BD2] stood out: its corresponding residue in BRD2 and BRD4 is glycine (Fig. 4d). Interestingly, these glycine residues were previously shown to be drivers of opposite paralog selective ternary complex formation with MZ1-VHL (Fig. 4d)[92]. Thus, we wondered if the corresponding glutamate residue in BRD3 might be responsible for the opposite paralog

selectivity with KLHDC2-SJ46421. Indeed, testing interactions with swap mutants by ITC showed that a E344G mutation of BRD3[BD2] effectively eliminated cooperative ternary complex formation ($\alpha = 1.7$; Figs. 4f, S4, Supplementary Table 6). Conversely, the G-to-E swap mutation of BRD2 and BRD4 promoted cooperative ternary complex formation ($\alpha = 3.2$ and 13 for BRD2[BD2] and BRD4[BD2] respectively; Figs. 4f, S4, Supplementary Table 6). Thus, the paralog

**Fig. 4 | SJ46421 promotes selective cooperative ternary complex formation with BRD3$^{BD2}$. a** Thermodynamic parameters for the indicated binary and ternary complexes. Data are the average +/− 1 s.d. from $n = 2$ experiments. Values for ΔG and ΔH are expressed in kcal/mol$^{-1}$. **b** Biochemical assay monitoring inhibition of KLHDC2 diGly protein substrate ubiquitylation by PROTAC-mediated ternary complex formation, performed as in Fig. 3f. SJ46421·BRD3$^{BD2}$ or SJ46421 alone (left panel) or SJ46423·BRD3$^{BD2}$ or SJ46423 alone (right panel) were incubated with neddylated CRL2$^{KLHDC2}$, prior to adding fluorescent ubiquitin-charged UBE2R2 (i.e. the pre-formed thioester-linked UBE2R2-ubiquitin intermediate) and diGly protein substrate. Effective PROTAC protein degraders promote ubiquitin transfer to BRD3$^{BD2}$ neo-substrate concomitant with loss of diGly substrate ubiquitylation. Samples were reduced when reaction was quenched by addition of SDS buffer. Loss of substrate ubiquitylation was quantified. Shown is representative panels from $n = 2$ independent experiments. **c** Quantification of loss of diGly substrate ubiquitylation from the gels in panel. **b** Data are the average from $n = 2$ independent

experiments. **d** Sequence alignment of the second bromodomain from BRD2, BRD3, and BRD4. Residue boundaries of the domains are shown, and E344 from BRD3$^{BD2}$ is highlighted in red (top panel). Superposition of BRD2$^{BD2}$ (salmon; 3ONI.pdb) and BRD3$^{BD2}$ (chocolate; 3S92.pdb) to the structure of the VHL-MZL-BRD4$^{BD2}$ ternary complex (cyan, orange, and red respectively;5T35.pdb). G382, E344, and G386 from BRD2, BRD3, and BRD4 respectively that mediate cooperative ternary complex formation with VHL are shown in sticks (bottom panel). **e** Electrostatic surface representation of KLHDC2 highlghting potential basic patches for interaction with BRD3 E344 (left panel). Cartoon representation from (**e**) with relevant basic residues shown in sphere representation (right panel). **f** Thermodynamic parameters for the indicated ternary complexes between KLHDC2 and the indicated swap mutant BRD proteins and the R56A mutant of KLHDC2. Experiments were performed $n = 2$ times. Source data are provided as a Source Data file.

selectivity is mediated largely by a single residue difference between BRDs.

Our discovery of BRD3's E344 promoting cooperative ternary complex formation prompted us to search for corresponding interacting residues in KLHDC2. Structural modelling approaches produced an ensemble of BRD3$^{BD2}$ binding conformations that were not informative. We thus started with an unbiased examination of the electrostatic potential of KLHDC2 (Fig. 4e). This revealed four basic patches centered around KLHDC2 R56, R112, K291, and H346/R347 (Fig. 4e). We mutated these residues to alanine and surveyed for their potential to cooperatively interact with BRD3$^{BD2}$. Of these, only the R56A mutation was able to eliminate cooperative ternary complex formation (α = 1.7; Figs. 4f, S4, Supplementary Table 6). Thus, R56 of KLHDC2 possess the features consistent with it complementing E344 of BRD3 to drive cooperative ternary complex formation.

## A PROTAC selectively employs KLHDC2 and overcomes its auto-inhibition

In PROTAC design, it is important to examine ubiquitin ligase selectivity, especially for E3s with close homologs with overlapping degron specificities. Several closely-related KLHDC-family proteins are CRL substrate receptors that all recognize C-degrons terminating in glycine[61,62]. Like CRL2$^{KLHDC2}$, the Kelch-type SBD of CRL5$^{KLHDC1}$ recognizes di-Gly degrons[73]. Meanwhile, two additional CUL2 based E3s, CRL2$^{KLHDC3}$ and CRL2$^{KLHDC10}$ prefer only slightly differing consensus sequences, K/R/Q-G and W/P/A-G, respectively[61,62]. As SJ46421 contains a C-terminal glycine-like moiety, we tested its E3 ligase selectivity in vitro (Figs. 5a, S5a). Of note, ubiquitylation by KLHDC3, KLHDC10, and KLHDC1 was poor (Figs. 5a, S5a). The data showed a greater than two orders-of-magnitude selectivity of SJ46421 for KLHDC2 over other KLHDC-family CRLs. Consistent with this view, SJ46411 and SJ46421 alone were only able to inhibit KLHDC2 substrate ubiquitylation and largely inactive towards other the Kelch family E3 ligases (Fig. S5b, c).

KLHDC2 is subject to auto-inhibition via its C-terminal Gly-Ser sequence mimicking a degron and mediating tetramerization (Fig. 5b)[72,79]. The distinctive form of regulation is facilitated by the degron mimic sequence at the C-terminus of one KLHDC2 binding the SBD of an adjacent protomer in the complex. Within the tetrameric assembly, all KLHDC2 substrate-binding pockets are occupied. As such, substrates must "capture" KLHDC2 monomers when they form transiently during the normal equilibrium between tetrameric and monomeric species (Fig. 5b). While this mechanism enables KLHDC2 to discriminate between bona-fide substrates and non-substrates, it is a unique feature for an E3 captured by heterobifunctional molecules. Specifically, ligand-directed ternary complex formation with CRL2$^{KLHDC2}$ would need to proceed with sufficient affinity and association kinetics to capture KLHDC2 monomers before they re-assemble into tetrameric species. If this does not occur - for example, in incubations with non-substrates for as long as 30 min - neddylated

CRL2$^{KLHDC2}$ tetramers mediate autoubiquitylation of the KHLDC2 subunit[79]. Thus, to assess the ability of SJ46421·BRD3 to effectively capture KLHDC2 monomers, we tested for a shift in substrate ubiquitylation from KLHDC2 to BRD3$^{BD2}$. Indeed, without SJ46421·BRD3$^{BD2}$, ubiquitin is exclusively conjugated to KLHDC2 (Fig. 5c, d). However, pre-incubations with SJ46421·BRD3$^{BD2}$ for as little as two minutes resulted in a complete switch in ubiquitin targeting to BRD3$^{BD2}$ (Fig. 5c, d). Thus, SJ46421·BRD3 possesses the characteristics of bona-fide KLHDC2 substrates.

## KLHDC2 PROTAC protein degraders promote selective BRD3 degradation in cells

Having successfully generated a heterobifunctional ligand that promotes high-affinity, cooperative, POI-selective ternary complexes with KLHDC2, which can overcome the intrinsic auto-inhibition of full-length KLHDC2, we next tested if our approach could target BRD protein degradation in cells. Since molecules with free acidic groups have less than ideal cellular permeability properties, we first compared the activities of the free acid ligand, SJ46421, to a methyl-ester variant, SJ46420 (Fig. 6a). A methyl-ester serves as a prodrug to enhance cellular permeability and can be converted to the free acid in cells by endogenous esterases. Although the free acid version SJ46421 was inactive in U2OS cells, its pro-drug variant SJ46420 efficiently and selectively targeted BRD3 for degradation, with partial reductions in the levels of BRD2 or BRD4 (Fig. 6b, Supplementary Table 7). This degradative activity was dependent on KLHDC2, as BRD3 levels were unaffected by SJ46420 treatment of a U2OS cell line with KLHDC2 knocked out (Fig. 6b, Supplementary Table 7). While SJ46420 was able to promote BRD3 degradation across numerous cell lines expressing endogenous KLHDC2, the efficiencies of targeting varied (Figs. 6b, S6, Supplementary Table 7).

We previously showed that KLHDC2 is expressed at very low levels in parental U2OS cells, and is -10-fold higher in a stable knockout/rescue cell line[79]. To determine the influence of KLHDC2 expression levels on degradative capacity, we performed our next set of experiments in this cell line. First, we tested various pro-drug moieties by comparing the activities of the methyl-ester, benzyl, and cyclopentyl-ester derivatives (Fig. 6a). While the cyclopentyl-ester derivative displayed activity much like that of the methyl-ester, the activity of the benzyl was significantly lower (Fig. 6c, Supplementary Table 7). The KLHDC2 knockout/rescue cell line maintained preferential PROTAC-targeted BRD3 degradation, with only partial downregulation of the levels of BRD2 and BRD4. Of note, the observed degradative capacity at 1 µM ligand in the KO/rescue line was equivalent to that of a 5 µM dose in the parental U2OS cell line. This suggested that increasing the levels of KLHDC2 enhances degradative capacity. We further validated CRL-dependence by co-dosing cells with SJ46420 and the NEDD8 E1 inhibitor MLN4924, which effectively stabilized BRD3 (Fig. 6d, Supplementary Table 7).

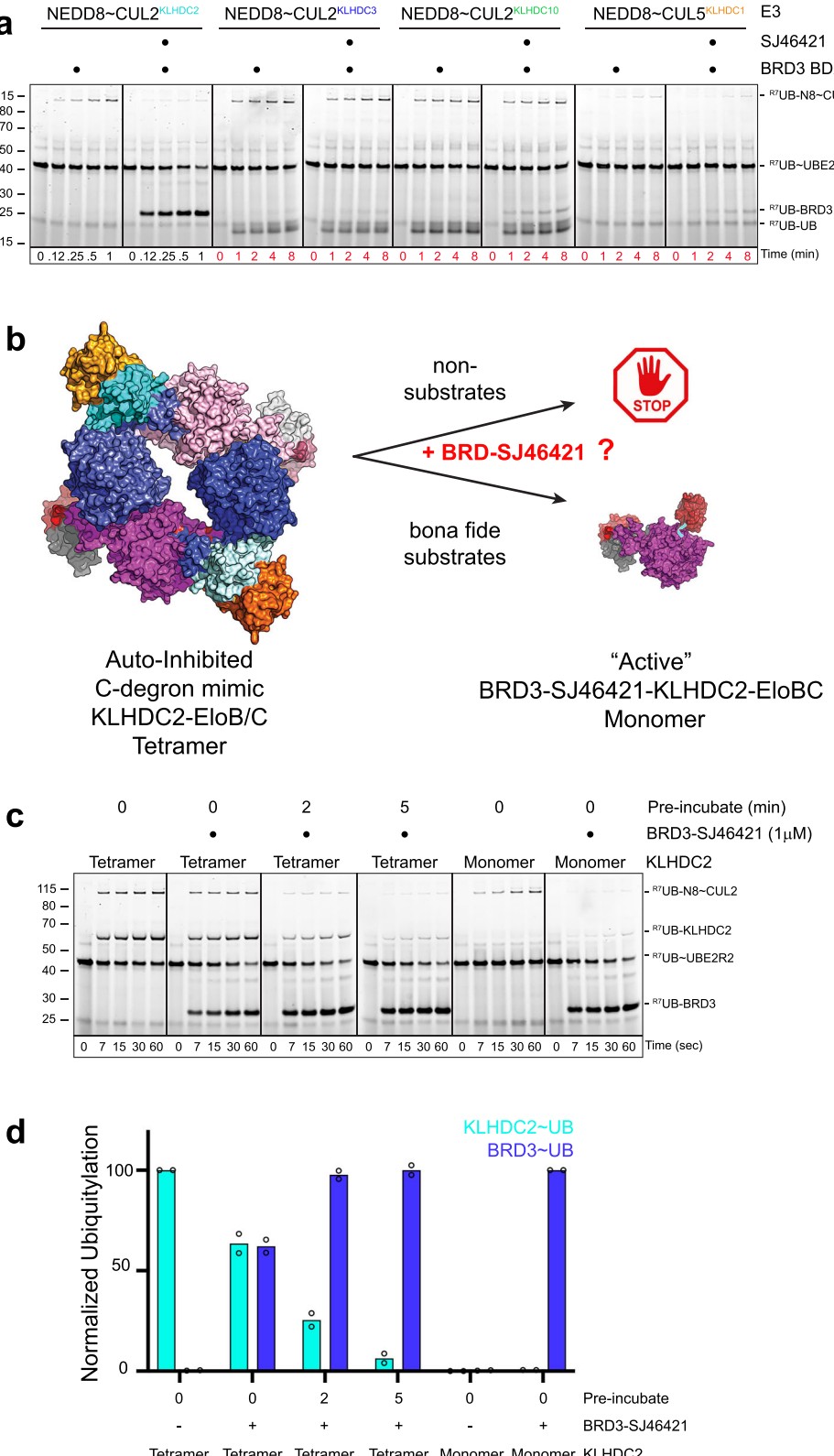

**Fig. 5 | KLHDC family selectivity and activity of SJ46421 towards tetrameric KLHDC2. a** Fluorescent scan of gel monitoring SJ46421-dependent BRD3$^{BD2}$ ubiquitylation by neddylated versions of CUL2$^{KLHDC2}$, CUL2$^{KLHDC3}$, CUL2$^{KLHDC10}$, or CUL5$^{KLHDC1}$. Assays were performed in pulse-chase format, monitoring PROTAC- and E3-dependent transfer of lysineless (R7) fluorescent ubiquitin from pre-formed thioester-linked UBE2R2-ubiquitin intermediate. Shown is representative panels from $n = 2$ independent experiments. **b** Cartoon depiction of the degron-mimic mediated auto-regulation of KLHDC2-EloBC by the inactive tetrameric assembly and active monomer-substrate complexes. **c** Fluorescent scan of ubiquitylation assays monitoring the ability of SJ46421-BRD3$^{BD2}$ to disrupt the KLHDC2-EloBC tetrameric assembly. Assays were performed as in (**a**). Shown is representative panels from $n = 2$ independent experiments. **d** Quantification of the levels ubiquitin conjugated KLHDC2 or BRD3$^{BD2}$ form gels in panel (**c**). Data are the average from $n = 2$ independent experiments. Source data are provided as a Source Data file.

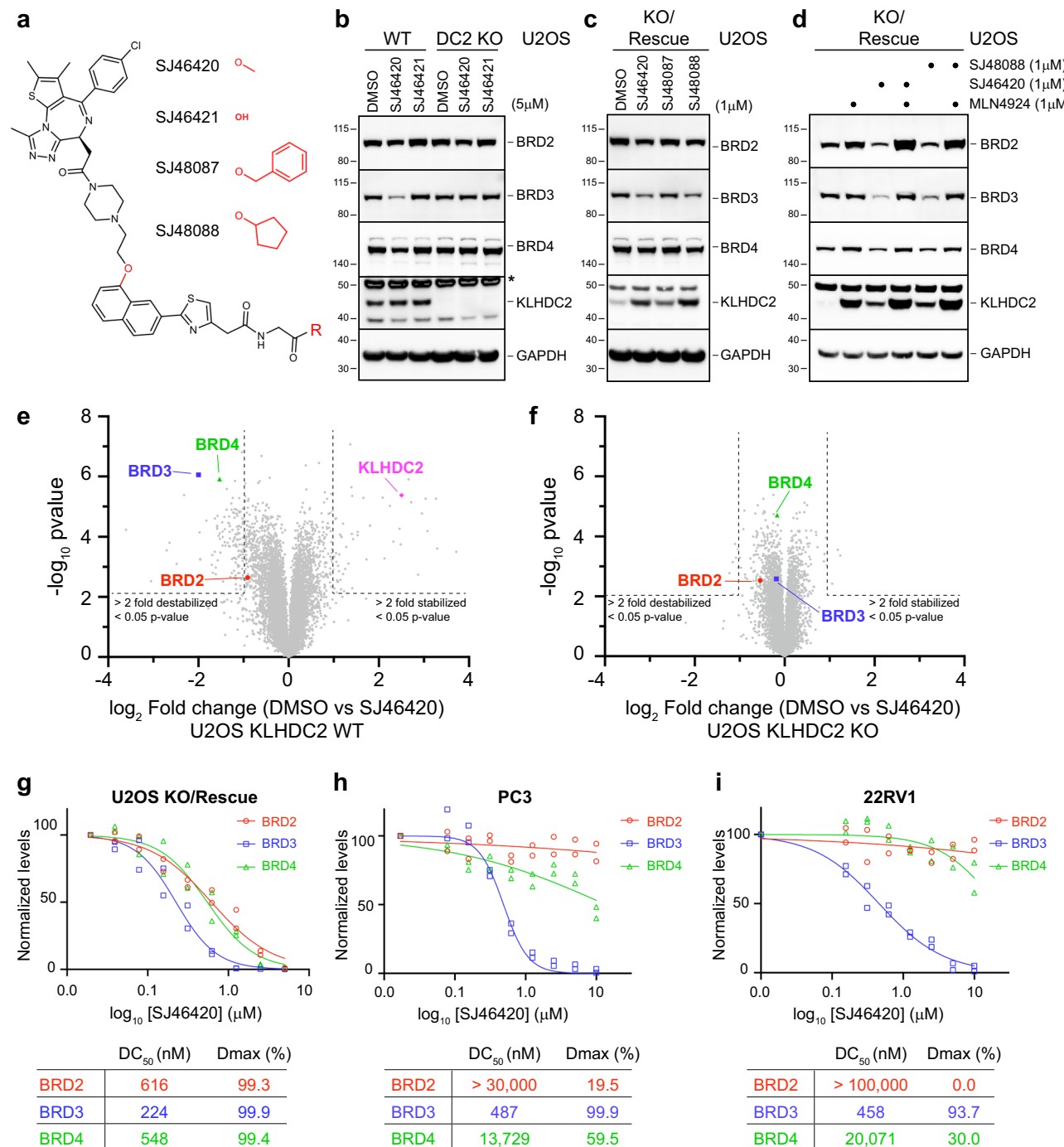

**Fig. 6 | KLHDC2 PROTAC protein degraders are active in cells and maintain BRD3 selectivity. a** Chemical structure of free-acid and pro-drug PROTAC variants of SJ46421. **b** Western blot monitoring the levels of the indicated proteins following a 24 h dose of wild-type U2OS or KLHDC2 knockout cells with DMSO, SJ46420, or SJ46421. The asterisk indicates a non-specific reactive band with the KLHDC2 antibody. **c** Same as (**b**) but with U2OS KLHDC2 KO/rescue cell line and the indicated pro-drug variants. **d** Same as (**c**) but with SJ46420 or SJ48088 in the absence or presence of co-dosing with 1 μM MLN4924. **e** Volcano plot from TMT proteomics comparing protein levels following a 24 h dose of U2OS KLHDC2 KO/rescue cell line with 1 μM SJ46420. Dashed lines demark proteins stabilized or destabilized > 2-fold

with *p* value < 0.05. Statistical tests used in the volcano plots is two-sided student's *t*-test and no correction was made for multiple comparisons. Data are from n = 3 biological replicates. **f** Same as (**e**) but with U2OS KLHDC2 knockout cells. **g** Quantification of dose-response western blot monitoring the levels of BRD2, BRD3, and BRD4 following a 24 h dose of U2OS KLHDC2 KO/rescue with the indicated concentration of SJ46420. DC$_{50}$ and D$_{max}$ values calculated from the fit shown in the table below are the average from *n* = 2 independent experiments. **h** Same as (**g**) but with PC3 cells. **i** Same as (**g**) but with 22Rv1 cells. All data in Fig. b-d are representative panels from *n* = 2 independent experiments. Source data are provided as a Source Data file.

To gain a more global view of selectivity and specificity, we performed TMT-based total proteomics on cells dosed with 1 μM SJ46420. Consistent with the results obtained by western blotting, in the KO/rescue U2OS cells, BRD3 was most substantially downregulated, with lesser reductions in the levels of BRD4 and BRD2 (Fig. 6e). These

SJ46420-dependent alterations to protein levels were not observed in the KO cells, and thus strictly depend on KLHDC2 (Fig. 6f).

Dose-response studies of SJ46420 were performed in a representative panel of cell lines to further characterize its efficacy, selectivity, and degradative capacity. The lowest DC$_{50}$ values were observed

with the U2OS KO/rescue line, perhaps owing to the relative over-expression of KLHDC2. These cells also showed Dmax values of ~100% for all BRDs, while maintaining ~2.5-fold selectivity for BRD3 over BRD2 and BRD4 (Figs. 6g, S6d, Supplementary Table 7). $DC_{50}$ values for BRD3 levels, of 487 nM and 458 nM, were measured from dosing of prostate cancer cell lines PC3 and 22Rv1. These cells maintained a high degree of selectivity for BRD3 over BRD4 (Figs. 6h, I, S6e, f, Supplementary Table 7, ~28 fold and ~44 fold in PC3 and 22Rv1 respectively). In these cell lines with endogenous KLHDC2, we were readily able to achieve $D_{max}$ values approaching 100% for elimination of BRD3.

We monitored the time-dependent degradation of BRDs by SJ46420 in SKBR3 and U2OS cells. Degradation of BRD3 was rapid in SKBR3 cells, reducing the levels of BRD3 by ~70% within two hours (Fig. S6g, Supplementary Table 7). Conversely, U2OS cells were somewhat delayed with maximal degradation occurring at 8 h (Fig. S6h, Supplementary Table 7). These time-course experiments suggested that SJ46420 was relatively unstable in cells as the levels of BRD3 seemed to rebound between 8 h and 24 h time points (Supplementary Table 7). These observations prompted us to perform LC-MS-MS studies with SKBR3 cells dosed at variable times with 10 μM SJ46420 or SJ46421 to quantitively measure the cellular concentration of SJ46421. As expected, the free acid SJ46421, possesses poor permeability properties with only minimal levels detected in cell lysates (Supplementary Table 8, max concentration 11 nM at 24 h). However, SJ46420 rapidly accumulated in cells at 4 h (Supplementary Table 8, 7227 nM at 4 h) and trailed off thereafter (Supplementary Table 8). Thus, future work will need to address SJ46420 metabolism to increase cellular half-life.

Finally, in an attempt to benchmark SJ46420, we compared its activity to the KLHDC2 ligand-/JQ1-based PROTAC protein degraders, K2-B4-3e and K2-B4-5e, recently reported by Arvinas. K2-B4-3e is the prodrug whose free acid counterpart K2-Be-3 was studied biochemically. The biochemical properties of K2-B4-5e were not described, but this showed superior efficacy relative to K2-B4-3e in cells. Comparing effects of treating SKBR3 cells for 4 h, or U2OS cells for 8 h, showed SJ46420 was 2-3 fold more potent than K2-B4-3e, while K2-B5-5e was 3-4 fold more potent than SJ46420 (Fig S6i, j, Supplementary Table 7). These experiments showcase the complementarity of differing approaches to identify degraders co-opting KLHDC2.

## Discussion

Herein, we describe the development of a PROTAC cooperatively employing KLHDC2, with target selectivity in vitro and across multiple cell lines. Thus, our study expands on recent work proposing the tractability of KLHDC2 for TPD[69–72]. Interestingly, ligands occupying the substrate-binding site have recently been obtained by distinct approaches. Two other groups used a computer-aided drug discovery approach, based on high-resolution E3-ligand structures[72,93]. Our study uniquely used a thermal-shift screen with a relatively small compound library. Collectively, the data show that both computational and experimental methods can successfully identify ligands as suitable starting points for employing a C-degron E3 for TPD.

Our study directly addresses several concepts for harnessing KLHDC2, and more broadly C-degron E3 ligases, for POI degradation by uniquely considering: (1) exit vector from the E3-binding ligand to achieve a cooperative E3-PROTAC-POI ternary complex; (2) neo-substrate selectivity from amongst a family of proteins engaged by the POI-binding ligand; (3) E3 ligase selectivity from amongst a family of proteins recognizing overlapping C-degron features; and (4) optimal prodrug options for achieving cell permeability.

Seminal work from the Ciulli lab demonstrated distinct correlation between the ability of PROTAC protein degraders to promote cooperative ternary complex formation and their efficiency and selectivity of degradation in cells[54,81,87,92,94]. Interestingly, the recently reported alternative PROTAC co-opting KLHDC2 did not promote cooperative ternary complex formation with a BRD target, and POI

selectivity was not investigated[72]. Inspection of the probable exit vectors of this KDRLKZ-1, and the two test cases developed in our study, indicated that SJ46421 (prodrug version SJ46420) would uniquely possess an exit vector similar to that of natural substrates (Fig. 3c). Accordingly, our PROTAC, SJ46421, promotes a highly cooperative and selective ternary complex with BRD3[BD2] (Fig. 4a). While future work is required to understand the structural basis of the selective and cooperative KLHDC2-SJ46421-BRD3 interaction, we note that repositioning the linker exiting from the KLHDC2 ligand by merely a single atom substantially reduced the cooperativity of ternary complex formation. Thus, our results underscore the importance of considering a multitude of possible exit vectors[57,95,96] during design of PROTAC protein degraders, including those harnessing C-end E3s.

The extreme C-terminal carboxyl group is a primary determinant of substrate recognition by KLHDC2[61,62,66]. Indeed, two other recently reported ligands[72,93], and a substantial percentage of hits from our biophysical screen, retain this feature. Such acid moieties in small molecules are notoriously prohibitive for cell permeability, and thus it was unknown if KLHDC2-targeting ligands could function in a cellular context. Furthermore, there is growing interest in developing small molecules mimicking protein-protein interactions involving protein C-termini[97]. Here, this limitation was overcome by utilizing prodrug versions of the ligands that overcome permeability issues and can be converted to their active counterparts by cellular esterases[98]. We further note that the choice of prodrug was impactful on activity of the PROTAC developed herein (Fig. 6c). As it is not clear if differences in prodrug effectiveness are due to permeability or impacting the activity of cellular esterases, we suggest future endeavors developing ligands towards C-degron E3s should explore several variants to discover those prodrugs with greatest activity.

KLHDC2 is but one member of a family of Kelch domain E3s that recognize substrate C-terminal glycine residues[61,62]. While structures showing substrate binding to KLHDC3, KLHDC10, and KLHDC1 remain to be reported, AlphaFold models suggest that many of the principles of this recognition are conserved[76,99]. While systemwide studies showed this family of E3 ligases can discriminate based on a substrate's penultimate residue[61,62], examination of specific substrates showed remarkable plasticity accommodating variant C-terminal sequences[74,76,79]. Thus, an important question was if a ligand targeting the C-terminal binding motif would be able to selectively bind a specific E3. This question would not be addressed in cell-based assays without first showing that the entire suite of C-degron E3s are active in that setting. Thus, employing reconstituted biochemical assays testing active E3 complexes side-by-side allowed us to show SJ46421 exhibits greater than two orders-of-magnitude selectivity for KLHDC2 over related CRLs (Figs. 5a, S5a). This systematic approach demonstrates the possibility of employing selective C-degron E3 ligases for further expansion of the E3 landscape for TPD.

One aspect of targeting E3s by ligands is the necessity to efficiently capture active E3s within the cellular milieu. Several recent studies have highlighted potential for nuanced regulation of some CRL family members. For example, some substrate receptors are subjected to autoubiquitylation and degradation in the absence of a bound substrate or neo-substrate[100,101]. In addition, many CRLs have capacity to form inactive oligomers that are overcome by substrate binding and neddylation[79,102–104]. The functional role of such autoinhibited oligomerization has been best demonstrated for KLHDC2, which amplifies substrate fidelity and activation of the substrate-bound E3. The tetramer:monomer (note that here, oligomerization status refers to the KLHDC2-EloB-EloC complex) equilibrium requires a bona-fide substrate to have a sufficiently fast on-rate[79]. By contrast, slower binding but equally high-affinity interactors that terminate with a KLHDC2[SBD]-binding di-Gly sequence are excluded[79]. Bonafide substrates drive E3 ligase activity in a feed-forward manner: to our knowledge, CRL2[KLHDC2] is the only CRL characterized to date where the substrate-bound

substrate receptor directly stimulates neddylation beyond simply promoting oligomer dissociation[79]. However, if the tetrameric (i.e. substrate-free) CRL2[KLHDC2] complex is neddylated, its KLHDC2 subunit is subject to autoubiquitylation and autodegradation[78,79]. Thus, if a PROTAC overcomes autoinhibition and enables target engagement by an active monomeric CRL2[KLHDC2], it also has potential to stabilize this E3. We, and a recent study[72], demonstrate biochemically that ternary complexes promoted by PROTAC protein degraders can efficiently convert the KLHDC2 tetrameric assembly into an active monomeric complex, and promote target degradation in cells expressing WT KLHDC2 (Figs. 5c, d, 6, S6). Notably, our assays also show that the levels of KLHDC2 increase upon treatment with an active PROTAC, especially in the U2OS KO/rescue line. If this reflects protection from autodegradation[78,101], then by extrapolation the particularly low amount of endogenous KLHDC2 may reflect limited endogenous utilization of this E3.

In summary, taken together with recent work, our studies demonstrate the utility of harnessing KLHDC2 for TPD. Furthermore, consideration of linkerology and prodrug moieties provides a blueprint for generating a cooperative, cell-active, and E3- and POI-selective PROTAC employing a C-degron E3.

## Methods

### Cell lines and cell culture
HEK293T (CRL-1573), MDA-MB-468 (HTB-132), U2OS (HTB-96), PC3 (CRL-1435), 22Rv1 (CRL-2505), AU565 (CRL-2351), and SKBR3 (HTB-30) cells were purchased from the American Type Culture Collection (ATCC). The KLHDC2 U2OS CRISPR KO and FLAG-KLHDC2 rescue cell lines were previously described. HEK293T and MDA-MB-468 cells were cultured in DMEM supplemented with 10% fetal bovine serum at 37 °C with 5% CO$_2$. U2OS and AU565 cell lines were cultured in McCoys 5 A supplemented with 10% fetal bovine serum at 37 °C with 5 % CO$_2$. PC3 cells were cultured in F-12K supplemented with 10% fetal bovine serum at 37 °C with 5 % CO$_2$. 22Rv1 and SKBR3 cells were cultured in RPMI-1640 supplemented with 10% fetal bovine serum at 37 °C with 5% CO$_2$. All cell lines were routinely checked for mycoplasma contamination with LookOut mycoplasma PCR detection kit (Sigma).

### Constructs
Expression constructs generated for this study were prepared by standard molecular biology techniques and coding sequences entirely verified. Mutant versions used in this study were generated by QuickChange (Stratagene). Constructs for bacterial expression of KLHDC2[SBD], AviTag- KLHDC2[SBD], UBE2R2, the NEDD8 E1 APPBP1-UBA3, UBE2M, UB, NEDD8, CUL2-RBX1, and CUL5-RBX2 were previously described. cDNA constructs for BRD2, BRD3, and BRD4 were obtained from Open Biosystems. Expression constructs were generated by cloning of BRD2[BD2] (residues 344-455), BRD3 [BD2] (residues 306-416), and BRD4[BD2] (residues 334-460) fragments into a pGEX-TEV based vector. Clones for insect cell expression of full-length KLHDC2-EloB/C, monomeric KLHDC2[KK]- EloB/C, full-length KLHDC3-EloB/C, KLHDC10-EloB/C, and CUL2-RBX1 were previously described. For expression of full-length KLHDC1-EloB/C, KLHDC1 was first cloned into pLib via Gibson assembly. Cassettes were then generated via PCR as described (Weissman) and Gibson assembled into pBig1a to generate a single vector for co-expression of all components. Proper assembly into pBig1a was confirmed by PmeI and SwaI restriction digestion.

### Protein expression and purification
UBE2M, UBE2R2, the NEDD8 E1 APPBP1-UBA3, were expressed in *E. coli* BL21 Gold (DE3) cells as GST fusion proteins. Fusion proteins were purified from cell lysates by glutathione affinity chromatography and liberated from GST by thrombin or TEV cleavage overnight at 4 °C. Cleavage reactions were further purified by ion-exchange and size exclusion chromatography in 25 mM HEPES, 200 mM NaCl, 1 mM DTT

pH = 7.5 (Buffer A). NEDD8, UB, BRD2[BD2], BRD3[BD2], and BRD4[BD2] were expressed in *E. coli* BL21 Gold (DE3) cells as GST- fusion proteins. Fusion proteins were purified from cell lysates by glutathione affinity chromatography and liberated from GST by thrombin or TEV cleavage during extensive dialysis overnight in Buffer A at 4 °C. Cleavage reactions were passed back over a glutathione affinity resin to remove free GST and any remaining uncleaved GST-fusion protein. Protein collected in the flow fraction was concentrated with an Amicon Ultra filtration unit and further purified by size exclusion chromatography in Buffer A.

KLHDC2[SBD] (15-361) and its Avi-tagged counterpart were expressed in *E. coli* Rosetta2 cells, grown in Terrific broth, with a N-terminal 6XHis-MBP-TEV tag. Protein was purified from cell lysates by Ni affinity chromatography. The 6XHis-MBP tag was liberated by cleavage with TEV overnight at 4 °C. Cleavage products were further purified by ion-exchange and size exclusion chromatography in Buffer A.

Full-length CUL2-RBX1 was co-expressed in insect cells as a His-Dac-TEV-CUL2 fusion protein with untagged RBX1. Full length KLHDC2-EloB/C, KLHDC3-EloB/C, KLHDC10-EloB/C, KLHDC1-EloB/C and mutant variants therof were expressed in insect cells with a N-terminal 6XHis tag on EloC. Proteins were purified from cell lysates by Ni affinity chromatography. Following TEV cleavage overnight at 4 °C cleavage reactions were further purified by ion exchange and size exclusion chromatography in Buffer A.

Full-length CUL5-RBX2 was co-expressed in insect cells as a GST-TEV-RBX1 fusion protein with untagged CUL5. Fusion proteins were purified from cell lysates by glutathione affinity chromatography and liberated from GST by TEV cleavage overnight at 4 °C. Cleavage reactions were further purified by ion-exchange and size exclusion chromatography in buffer A.

Neddylation and purification of CUL2–RBX1 and CUL5-RBX1, were prepared as described previously for CUL1-RBX1. Briefly, the final concentrations of components in the neddylation reactions were as follows: 12 μM CUL2-RBX1, 1 μM UBE2M, 0.1 μM APPBP1-UBA3, and 20 μM NEDD8 in 25 mM HEPES, 200 mM NaCl, 10 mM MgCl$_2$, 1 mM ATP, pH = 7.5. CUL5-RBX2 reactions contained 1 μM UBE2F in place of UBE2M. Reactions were initiated at room temperature by the addition of NEDD8 and incubated for ten minutes prior to quenching by the addition of DTT to 10 mM. Quenched reactions were spun at 17 K x g for 10 min and immediately applied to a Superdex SD200 column to purify NEDD8-CUL2-RBX away from reaction components.

### Protein modifications
Conditions for the Biotinylation of Avi-KLHDC2[SBD] were previously described. Briefly, reactions contained 50 μM Avi-KLHDC2 and 1.5 μM BIRA in 10 mM Tris, 10 mM ATP, 10 mM magnesium acetate, 2 mM Biotin, pH 8.0 and were incubated at room temperature for 1 h and continued overnight at 4 °C. The reaction mixture was diluted fourfold into 25 mM Tris, 1 mM DTT, pH 8.0 and purified over a ion-exchange column. Fractions containing biotin-KLHDC2 were pooled, concentrated, and further purified by size exclusion chromatography in 25 mM HEPES, 200 mM NaCl, 1 mM DTT, pH 7.5.

To introduce a cysteine for fluorescent labeling of UB and K0UB we mutated the protein kinase a site in the pGEX2TK backbone converting the PKA site from RRASV to RRACV. NEDD8, UB or UBK0 purified from this expression construct were labeled with AlexaFluor 488 Maleimide or Fluorescein-5-Maleimide respectively as previously described[49]. Briefly, DTT was added to UB or UBK0 at a final concentration of 10 mM and incubated on ice for 20 min to completely reduce cysteines for labeling. DTT was removed by buffer exchange over a NAP-5 column (GE Healthcare) in labeling buffer (25 mM HEPES, 200 mM NaCl). Labeling reactions consisted of UB or UBK0 at 150 μM final concentration and were initiated by the addition of 600 μM AlexaFluor 488 Maleimide or Tetramethylrhodamine-5-Maleimide (4X excess over labeling target and <5% final DMSO concentration). Reactions were incubated at room

temperature for 2 h and quenched by the addition of DTT to 10 mM. Quenched reactions were desalted over a PD-10 column in labeling buffer containing 1 mM DTT to remove unreacted probe. Desalted protein was concentrated in an Amicon Ultra filtration unit and further purified over a Sephadex SD75 column.

## Cell biological experiments

Dosing experiments with U2OS cells and its variants were performed by seeding $4.5 \times 10^5$ cells into 60 mM dishes on day one. Fresh media was added on day 2 and day 3, and cells dosed with vehicle or the indicated concentration of test compounds. Cells were harvested at the indicated time points and processed for western blotting.

Dosing experiments with HEK293T, MDA-MD-468, PC3, 22Rv1, AU565, and SKBR3 cells were performed by seeding $8.0 \times 10^5$ cells into 6 well plates on day one. On day 2 fresh media was added and cells dosed with vehicle or the indicated concentration of test compounds. Cells were harvested at the indicated time points and processed for western blotting.

Samples for western blots were prepared by resuspension of cell pellets in lysis buffer consisting of 50 mM Tris, 150 mM NaCl, 0.5% NP-50, 6 M Urea, 0.1 % SDS, 1X HALT protease and phosphatase inhibitor cocktail, 250 U universal nuclease, pH 7.5. Samples were lysed on ice for 15 min and cellular debris cleared by a 10 min spin at 17 K x g. The protein concentration of supernatants was determined by BCA protein assay. Equivalent total amounts of protein lysate were mixed with 2X SDS-PAGE sample buffer, separated on 4–12 % Bis-Tris gels, transferred to PVDF membranes, and probed with the indicated antibodies. Western blots were developed with SuperSignal West Pico Plus substrate and chemiluminescent signal read on an ImageQuant LAS4000. Western blot images were obtained through detection of BRD2 (Cell Signaling D89b4 1:1000), BRD3 (Santa Cruz sc-81202 1:500), BRD4 (Cell Signaling E2A7X 1:2000), KLHDC2 (Atlas Antibodies HPA000628 1:1000), FLAG (Sigma F1804 1:1000), and GAPDH (Santa Cruz sc-32233 1:4000). Western blots were quantified using ImageJ (version 1.53 k). For an example of presentation of full scan blots, see the Source Data file.

## TMT proteomics

**Sample preparation.** Cell pellets were resuspended in lysis buffer (50 mM EPPS pH 8.0, 100 mM NaCl, 2% SDS) supplemented with protease inhibitor tablets (Sigma) and further lysed by passing lysate through a 25 G needle for 10–15 strokes. Lysates were then centrifuged at 17,000 $g$ for 5 min at room temperature. Protein concentrations were measured using a BCA kit and 120 μg of protein extract was transferred to a protein lo-bind tube for further processing. Each sample was reduced with 5 mM TCEP at room temperature for 30 min and subsequently alkylated with 15 mM iodoacetamide for 30 min at room temperature, covered in foil. Reaction was quenched with 10 mM DTT for 15 min at room temperature, covered in foil. Methanol-chloroform precipitation was used to precipitate proteins: four parts methanol was added and vortexed, one part chloroform was added and vortexed, and three parts water was added and vortexed. Sample with protein precipitate was then centrifuged for 5 min at room temperature at 14,000 $g$. Upper aqueous layer was carefully aspirated, and methanol was added and vortexed. Sample was centrifuged at 21,000 $g$ for 5 min and a second round of methanol wash was performed. Sample was centrifuged at 21000 $g$ for 5 min and supernatant was carefully aspirated without disturbing the protein precipitate pellet. Remaining methanol was dried using a vacuum centrifugation. Dried protein pellet was resuspended in 100 μl of 200 mM EPPS pH 8.0 and digested with LysC (1:50) at 25 °C overnight. Trypsin (1:50) was further added and incubated at 37 °C for 6 h. Post-digestion, peptide was quantified using a colorimetric peptide quantification kit (Thermo Fisher) and 100 μg of peptides was used for tandem mass tag (TMT) labeling. Experiments were performed in biological triplicate.

**TMT-labeling.** 100 μg of peptides was labeled with 150 μg of TMT reagent, in a solution with 30% (v/v) acetonitrile (ACN), for 1.5 h at room temperature, with shaking on a thermomixer. Labeling reaction was quenched with 0.5% (v/v) hydroxylamine for 15 min at room temperature. Labeled samples were pooled 1:1 to ensure sample amount of peptide was represented in each channel. Pooled sample was dried in a vacuum centrifuge and the resulting pellet was resuspended in 1 ml of 2.5% formic acid. Resuspended sample was desalted using a C18 solid-phase extraction (SPE) column (200 mg, Sep-Pak, Waters) and dried in a vacuum centrifuge.

**Off-line basic pH reversed-phase (BPRP) fractionation.** We fractionated the pooled, labeled peptide sample using BPRP HPLC[105] and an Agilent 1260 pump equipped with a degasser and a UV detector (set at 220 and 280 nm wavelength). Peptides were subjected to a 50 min linear gradient from 5% to 35% acetonitrile in 10 mM ammonium bicarbonate pH 8 at a flow rate of 0.6 mL/min over an Agilent 300Extend C18 column (3.5 μm particles, 4.6 mm ID and 250 mm in length). The peptide mixture was fractionated into a total of 96 fractions, which were consolidated into 24 super-fractions[106], all of which were analyzed. Samples were subsequently acidified with 1% formic acid and vacuum centrifuged to near dryness. Each super-fraction was desalted via StageTip, dried again via vacuum centrifugation, and reconstituted in 5% acetonitrile, 5% formic acid for LC-MS/MS processing.

**Liquid chromatography and tandem mass spectrometry.** Mass spectrometric data were collected on Orbitrap Fusion Lumos instruments coupled to a Proxeon NanoLC-1200 UHPLC. The 100 μm capillary column was packed with 35 cm of Accucore 150 resin (2.6 μm, 150 Å; ThermoFisher Scientific) at a flow rate of ~400 nL/min. The scan sequence began with an MS1 spectrum (Orbitrap analysis, resolution 60,000, 400 – 1600 Th, automatic gain control (AGC) target 100%, maximum injection time "auto"). Data were acquired ~90 min per fraction. MS2 analysis consisted of collision-induced dissociation (CID), quadrupole ion trap analysis, automatic gain control (AGC) 100%, NCE (normalized collision energy) 35, $q$-value 0.25, maximum injection time 35 ms), and isolation window at 0.7 Th. RTS was enabled and quantitative SPS-MS3 scans (resolution of 50,000; AGC target $2.5 \times 10^5$; collision energy HCD at 55%, max injection time of 250 ms) were processed through Orbiter with a real-time false discovery rate filter implementing a modified linear discriminant analysis. For FAIMS, the dispersion voltage (DV) was set at 5000 V, the compensation voltages (CVs) used were −40 V, −60 V, and −80 V and the TopSpeed parameter was set at 1 s.

**Data analysis.** Spectra were converted to mzXML via MSconvert[107]. Database searching included all entries from the human UniProt reference Database. The database was concatenated with one composed of all protein sequences for that database in reversed order. Searches were performed using a 50 ppm precursor ion tolerance for total protein level profiling. The product ion tolerance was set to 0.9 Da. These wide mass tolerance windows were chosen to maximize sensitivity in conjunction with Comet searches and linear discriminant analysis[108,109]. TMTpro labels on lysine residues and peptide N-termini +304.207 Da), as well as carbamidomethylation of cysteine residues (+ 57.021 Da) were set as static modifications, while oxidation of methionine residues (+ 15.995 Da) was set as a variable modification. Peptide-spectrum matches (PSMs) were adjusted to a 1% false discovery rate (FDR)[110,111]. PSM filtering was performed using a linear discriminant analysis, as described previously[109] and then assembled further to a final protein-level FDR of 1%[111]. Proteins were quantified by summing reporter ion counts across all matching PSMs, also as described previously[112]. Reporter ion intensities were adjusted to correct for the isotopic impurities of the different TMTpro reagents

according to manufacturer specifications. The signal-to-noise (S/N) measurements of peptides assigned to each protein were summed and these values were normalized so that the sum of the signal for all proteins in each channel was equivalent to account for equal protein loading. Finally, each protein abundance measurement was scaled, such that the summed signal-to-noise for that protein across all channels equals 100, thereby generating a relative abundance (RA) measurement.

### TR-FRET assay
TR-FRET assays were carried out in black 384-well microtiter plates at a final volume of 20 μL per well. An assay cocktail was prepared as a mixture of 50 nM Biotin-KLHDC2 SBD, 20 nM AlexaFluor488-RIC8B peptide, 2.5 nM Tb-Streptavidin (ThermoFisher) in assay buffer (25 mM HEPES, 100 mM NaCl, 0.1% Triton X-100, 0.5 mM DTT, pH 7.5). The assay cocktail was incubated for thirty-minutes at room temperature and 20 μl distributed to assay plates. Molecules were distributed to the plates at the indicated concentration and the assay mixture was incubated for 1 h at room temperature prior to measuring the TR-FRET signal with a PHERAstar FS plate reader (BMG Labtech) equipped with modules for excitation at 337 nm and emissions at 490 and 520 nm. The integration start was set to 100 μs and the integration time to 200 μs. The number of flashes was fixed at 100. The ratio of 520/490 was used as TR-FRET signal in calculations. Assay endpoints were normalized to 100% (buffer only) and used for curve fitting in Prism.

### Crystallography
For the structures of KLHDC2$^{SBD}$ bound to small molecules, KLHDC2 (250 μM final concentration) was crystallized at room temperature in 13–18% PEG3350, 0.2 M NaSCN, 0.1 M Bis-Tris Propane, pH = 7.5. Crystals were soaked in mother liquor supplemented with 1 mM small molecule for 24–48 h. Crystals were harvested in mother liquor supplemented with 25% MPD prior to flash-freezing in liquid nitrogen. Reflection data were collected at Sercat 24-ID at the Advanced Photon Source and AMX-17-ID-1 at Brookhaven National Laboratory. The crystals belong to space group P2$_1$ with two KLHDC2-small molecule complexes in the asymmetric unit. Phases were obtained by molecular replacement using PHASER[113] searching for two copies of KLHDC2 (with peptide omitted) from 6DO2.pdb. Manual building was performed in COOT and refinement was performed using Phenix[114].

### Biochemical assays
The use of pulse-chase assays allowed comparing the paths of UB transfer starting from UBE2R2. First, UBE2R2 was pulse-labeled by incubating a mixture of UBA1 (0.3 μM), E2 (10 μM), and fluorescently labeled UB or KOUB (15 μM) in 25 mM HEPES, 100 mM NaCl, 100 mM MgCl$_2$, ATP (2 mM), pH 7.5 at room temperature for 13 min. Pulse-loading reactions were quenched by the addition of EDTA to 50 mM and incubated on ice for 5 min. Ubiquitylatioin chase reactions consisted of mixing the E2 ~ *UB thioester conjugate (0.4 μM final concentration) with the indicated pre-equilibrated NEDD8 ~ CUL2-RBX1-KLHDC2-EloB/C, NEDD8 ~ CUL2-RBX1-KLHDC3-EloB/C, NEDD8 ~ CUL2-RBX1-KLHDC10-EloB/C, or NEDD8 ~ CUL5-RBX2-KLHDC1-EloB/C (0.3 μM final concentration) with or without the indicated concentrations of PROTAC-BRD$^{BD2}$ in 25 mM HEPES, 100 mM NaCl, 50 mM EDTA, 0.5 mg/ml BSA, pH 7.5 at room temperature. Reactions were quenched at the indicated times with 2X SDS-PAGE sample buffer and separated on 4–12% Bis-Tris gradient gels, scanned for fluorescence on a Typhoon imager. All experiments, except where indicated in Fig. 5c utilized monomeric KLHDC2-EloB/C. Experiments monitoring the disruption of the KLHDC2-EloB/C tetramer were performed as described above, except NEDD8 ~ CUL2-RBX1-KLHDC2-EloB/C was pre-incubated with SJ46421-BRD3$^{BD2}$ (0.3 μM final concentration) for the indicated times prior to starting ubiquitylation reactions by the addition of UBE2R2 ~ K0UB.

To monitor small molecule or ternary complex inhibition of KLHDC2 substrate ubiquitylation we used a modified pulse-chase format. We used the artificial KLHDC2 substrate NEDD8Ins2, which contains a two amino acid insertion in the tail of NEDD8. We chose to utilize this substrate due to its rapid association/dissociation kinetics with KLHDC2 (K$_d$ = 355 nM). In addition, Lys48 of NEDD8 is targeted with rapid kinetics via UBE2R2. UBE2R2 was pulse-loaded with fluorescent UB as described above. Chase reactions consisted of the simultaneous addition of UBE2R2 ~ *UB thioester conjugate (0.2 μM final concentration) and NEDD8Ins2 (0.6 μM final concentration) with NEDD8 ~ CUL2-RBX1-KLHDC2-EloB/C (0.02 μM final concentration) pre-equilibrated with the indicated concentrations small molecule or PROTAC-BRD3$^{BD2}$. Reactions were quenched at the indicated times with 2X SDS-PAGE sample buffer and separated on 4-12% Bis-Tris gradient gels, scanned for fluorescence on a Typhoon imager.

### Thermal shift assay
Experimental compounds (200 nl each) were dispensed into a 384-well PCR plate using the Labcyte ECHO 655 T. The assay solution, comprising 0.5 μM KLHDC2, 25 mM HEPES, 200 mM NaCl, and 5% Sypro Orange at pH 7.5, was added to achieve a final volume of 20 μl. Negative control wells contained only DMSO, while positive control wells contained a di-Gly substrate peptide derived from SELK. Fluorescence measurements of Sypro Orange in each sample were taken using a Quantstudio 5 or Quantstudio 6, with temperatures ranging from 23 to 75 °C at a rate of 0.05 °C/s. Data analysis involved calculating thermal stability using the Boltzmann and first derivative minima equations in Thermo Scientific Protein Thermal Shift software v1.4. The resulting data were uploaded to CDD Vault for further analysis.

The 8892-compound lead-like one library was screened for its ability to stabilize KLHDC2 using a sypro orange-based thermal stability single-point assay. Three percent of these compounds were cherrypicked for further analysis. These compounds were tested in triplicate, and seven were selected based on reproducibility and data quality (0.07% of the original screen). When tested in dose-response format, three of these compounds were eliminated based on data quality and potency of the ligands. The four remaining compounds gave strong dose-response data in the sypro orange-based thermal stability assay. When tested in a sypro orange independent thermal stability assay using the 350/330 nm absorbance over a temperature range, two further compounds were eliminated. These findings were further validated with TR-FRET EC$_{50}$ values of >200 μM for the four eliminated compounds. SJ6145 (our lead compound for this study) and SJ8152 were retained for further analysis. While SJ8152 showed a higher change in thermal stability, the potency of this compound was lower. Unlike SJ6145, the attempted optimization of SJ8152 did not improve the potency of this ligand.

### Isothermal titration calorimetry
All experiments were performed on a MicroCal AutoITC200. Proteins were first buffer matched by desalting over a NAP5 column in ITC buffer (25 mM HEPES, 150 mM NaCl, 1 mM BME, pH = 7.5). Protein concentrations were determined by nanodrop and diluted to the desired concentration with 1:10 dilution of ITC buffer supplemented with 0.5% Triton-X100, and DMSO as required to match the final concentration in small molecule dilutions. Experiments measuring small molecule binding to the KLHDC2$^{SBD}$ contained 8 μM final concentration of small molecule in the sample cell and 150 μM KLHDC2$^{SBD}$ in the syringe. Experiments measuring small molecule binding to BRD2$^{BD2}$, BRD3$^{BD2}$, or BRD4$^{BD2}$ contained 17 μM final concentration of small molecule in the sample cell and 385 μM BRD$^{BD2}$ in the syringe. Experiments measuring KLHDC2$^{SBD}$ ternary complex formation BRD2$^{BD2}$, BRD3$^{BD2}$, or BRD4$^{BD2}$ contained 8 μM final concentration of SJ46421 and 14 μM BRD$^{BD2}$ in the sample cell and 150 μM KLHDC2 in the syringe. Experiments with SJ46423 were as described above except the

final concentration of SJ46423 or SJ46423-BRD3$^{BD2}$ was 40 µM while the final concentration of KLHDC2$^{SBD}$ was 600 µM. Titrations were performed at 30 °C with one injection of 0.4 µl, followed by 13 injections of 3 µl.

## Surface plasmon resonance

**Method for SJ6145 and SJ10278**. Surface plasmon resonance (SPR) was utilized to determine kinetic parameters and binding affinity of KLHDC2 substrates using a PioneerFE (Sartorius). Experiments were carried out in 20 mM Tris, 200 mM NaCl, 1 mM TCEP, 0.01% Brij35, and 1% DMSO at pH 7.4 as a running buffer with a 100 µL/min flow rate at 20 °C.

The 10xHis-KLHDC2 was immobilized onto a Ni-NTA-coated biosensor chip (HisCap). The chip surface was conditioned using 10 mM HCl, 50 mM NaOH, and 0.1% SDS (50 µL at 100 µL/min) twice, followed by a 100 mM Tris buffer wash (100 µL at 100 µL/min). The surface was then activated using an injection of 0.5 M EDTA (125 µL at 25 µL/min) followed by an injection of 5 mM NiCl$_2$ (125 µL at 25 µL/min). Protein was then immobilized onto channel 3 at around 3500 RU. Channel 2 was treated identically to channel 3 except for protein loading to be used as a reference channel. The compounds (SJ6145 and SJ10278 in DMSO) were diluted in a non-DMSO buffer (20 mM Tris pH 7.4, 200 mM NaCl, 1 mM TCEP, 0.01% Brij35) to become 1% DMSO and injected using the OneStep® injection method on the Pioneer system which utilizes Taylor dispersion to obtain binding kinetics data from a single injection. 100% of the loop was injected with a 300 s dissociation time. All samples were run in triplicate. Data was analyzed using Qdat software and fit to a simple 1-site model.

**Method for SJ4611 and SJ46418.** The SPR experiments were performed to determine the binding affinities (K$_D$) of KLHDC2-SJ46411 and KLHDC2-SJ46418 complexes. All the experiments were performed using a Biacore 1S+ instrument. The experiments were conducted at a 30 µL/min flow rate and 25 °C temperature. The running buffer consisted of 20 mM Tris pH 7.5, 200 mM NaCl, 1 mM TCEP, 0.01% Brij35, and 4% DMSO. The Cytiva Biotin CAPture kit, Series S (289202234), was used to immobilize the biotinylated KLHDC2 to the sensor chip surface. The reference channel (channel 1) and active channel (channel 2) were conditioned and activated identically, and KLHDC2 was not introduced to the reference channel (channel 1). In brief, before the immobilization, the sensor chip surface was conditioned by injecting a mixture of (3:1 v/v) 8 M guanidine hydrochloride and 1 M NaOH at a flow rate of 5 µL/min for 2 min for two times. Then, running buffer was injected at a 30 µl/min flow rate for 1 min 3 times. Next, the Biotin CAPture reagent was injected at a flow rate of 2 µl/min for 5 min. Then, biotinylated KLHDC2 was only injected onto the active channel (channel 2).

To determine the affinity of KLHDC2- SJ46411, biotinylated KLHDC2 was immobilized to 2500 RU onto the active channel. Then, a concentration series of SJ46411 (0.15 µM to 9.4 µM) was prepared by diluting stock solutions of SJ46411 (which was dissolved in DMSO) in a non-DMSO buffer (20 mM Tris pH 7.5, 200 mM NaCl, 1 mM TCEP, 0.01% Brij35) to become 4% DMSO. In the KLHDC2- SJ46418 experiment, biotinylated KLHDC2 was immobilized to 2000 RU onto the active channel, and the concentration series of 0.098 µM to 12.5 µM was tested. Sample preparation was carried out precisely the same as SJ46411. All experiments were performed using a multi-cycle analysis method with a 30 µl/min flow rate, 120 S association, and 240 s dissociation time. All data were reference subtracted and corrected for DMSO mismatch. The sensorgrams were analyzed using Biacore insight evaluation software, and K$_D$ was determined using the 1:1 steady-state affinity fitting. Finally, all the analyzed data was replotted using GraphPad Prizm 10 software.

## Measurement of SJ46421 cellular concentration by LC-MS/MS

SKBR3 cells were dosed with 10 µM SJ46420 or SJ46421 and harvested at the indicated times. Cells were counted for normalization and washed 2X with 1X PBS. The final pellet was resuspended in 100 µL of acetonitrile

to denature the cells. The samples were then centrifuged at 16,100 rcf for 5 min and 75 µL of each supernatant was aliquoted into a 96-well analytical plate and mixed with 75 µL of a 50% acetonitrile solution (1:1 v/v) containing 100 nM of JQ1, which served as an internal standard. The samples were mixed thoroughly before being analyzed by LC-MS/MS.

Samples were analyzed using an Acquity UPLC (Waters Corporation) coupled with a 6500 Triple Quad System (AB Sciex). Briefly, 5 µL of extract was separated on an Acquity UPLC BEH C18 1.7 µm, 2.1 × 50 mm column, maintained at 60 °C. Mobile phase A was 0.1% formic acid (FA) in MilliQ water, and solvent B was 0.1% FA in acetonitrile. The inlet method for these samples used a flow rate of 0.8 mL/min with the following gradient: 0 – 0.4 min, 70.0% solvent A and 30% solvent B; 0.4–1.5 min, gradient of 70-5% solvent A and 30–95% solvent B; 1.5–1.8 min, hold on 5% solvent A and 95% solvent B; 1.8–1.95 min gradient of 5–70% solvent A and 95–30% solvent B; and 1.95–2 min, hold on 70% solvent A and 30% solvent B. The first 0.4 min of eluate was desalted to waste by an integrated Valco valve. Multiple reaction monitoring (MRM) was used to quantify the compound based on a quadratic calibration curve with a weighing of $1/x^2$. The mass spectrometer was operated in positive ion mode with electrospray ionization. Reaction monitoring parameters were as follows: the pressure of ion source gas 1 = 50 psi, the pressure of ion source gas 2 = 50 psi, the pressure of curtain gas = 30 psi, the pressure of collision gas = 8 psi, ion spray voltage = 5.5 kV, source temperature = 500 °C. The MRM transitions were m/z 837.27 to m/z 383.11 for SJ46421 and m/z 457.17 to m/z 401.02 for JQ1. The declustering potentials, entrance potential, collision energy, and collision cell exit potential were set at 50 V, 10 V, 55 V, and 12 V for SJ46421; and 100 V, 10 V, 19 V, and 13 V for JQ1. Data were acquired using Analyst Software (AB Sciex, version 1.6.3) and analyzed using MultiQuant Software (AB Sciex, version 3.0.3).

## Chemical synthesis

All synthetic methods and associated analytical data for the compounds used in this study, including SJ46421 and SJ46423, are provided in Supplementary Figures 7–25.

## Reporting summary

Further information on research design is available in the Nature Portfolio Reporting Summary linked to this article.

## Data availability

The X-ray crystallography data have been deposited in the RCSB with accession codes 9BCA (KLHDC2 bound to SJ10278), 9BC9 (KLHDC2 bound to SJ46411), and 9BCC (KLHDC2 bound to SJ46418). Proteomics data generated during the study are available via ProteomeXchange Consortium via the PRIDE[115] partner repository, under the dataset identifier PXD051581 (Effects on the proteome of U2OS KLHDC2 knockout and knockout/rescue cells for SJ46420). All other data generated for Tables, Figures, and Supplementary Figs. are available in the Source data file. Source data are provided with this paper.

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

## Acknowledgements

We thank the St. Jude Department of Chemical Biology and Therapeutics Chemistry, Lead Discovery Informatics, Project Management, and High Throughout Biosciences Centers, and the St. Jude Biomolecular X-Ray Crystallography Center for advice, assistance, and support. This research used resources of the Advanced Photon Source; a U. S. Department of Energy (DOE) Office of Science user facility operated for the DOE Office of Science by Argonne National Laboratory under Contract No. DE-AC02-06CH11357. The Center for BioMolecular Structure (CBMS) AMX, is primarily supported by the National Institutes of Health, National Institute of General Medical Sciences (NIGMS) through a Center Core P30 Grant (P30GM133893), and by the DOE Office of Biological and Environmental Research (KP1607011). The National Synchrotron Light Source II is operated under Contract No. DE-SC0012704 for the U.S. Department of Energy (DOE), Office of Science. We thank Miklos Bekes, David R. Langley and Kurt Zimmermann and Arvinas for generously providing K2-B4-3e and K2-B5-5e. This research was supported by ALSAC, St. Jude Children's Research Hospital, NIH P30CA021765 to the St. Jude Children's Research Hospital Comprehensive Cancer Center, NCI 5RO1CA247365 (D.C.S. and B.A.S.), NIH R01GM132129 (JAP), NIH R01AG11085 (J.W.H. and S.J.E.), Howard Hughes Medical Institute (S.J.E.), and Max-Planck-Gesellschaft (B.A.S.). S.J.E. is an Investigator with the Howard Hughes Institute and a member of the Harvard Ludwig Institute.

## Author contributions

D.C.S., S.J.E., R.E.L., B.A.S. conceived the project. D.C.S. and M.T.K. generated protein complexes and performed their biochemical characterization. D.C.S. performed biochemical assays, determined crystal structures, and guided structure-based small molecule improvement. DJM collected X-ray crystallography data. E.G. performed and R.E.L. supervised the thermal shift screen. S.C.C. performed and T.C. supervised the TR-FRET assays. S.D., J.R., and R.T. performed small molecule chemical syntheses, supervised by R.E.L. C.T.G. and T.J. performed SPR. CL and JAP performed proteomics and analyzed the proteomics data, supervised by J.W.H. J.O. performed structural modeling. L.Y., Y.L., and V.L. performed permeability and stability studies. D.C.S. and H.W.L. performed cell biology. D.C.S., S.D., R.E.L and B.A.S. prepared the manuscript with input from all authors. R.E.L. and B.A.S. supervised the project.

## Funding

## Competing interests

D.C.S., S.D., E.G., S.C.C., J.R., R.T., C.T.G., H.W.L., J.O., T.C., R.E.L., and B.A.S. are listed as co-inventors in patent filings associated with the technologies described in this manuscript. D.C.S. and B.A.S. are co-inventors of intellectual property that is unrelated to this work (DCN1 inhibitors licensed to Cinsano). J.W.H. is a founder and consultant for Caraway Therapeutics and is a scientific advisory board member for Lyterian Therapeutics. S.J.E. is a founder of, and holds equity in, TScan Therapeutics and Immune ID. S.J.E. is also founder of MAZE Therapeutics, and Mirimus and serves on the scientific advisory board of TSCAN Therapeutics, and MAZE Therapeutics. In accordance with Partners HealthCare's conflict of interest policies, the Partners Office for Interactions with Industry has reviewed SJE's financial interest in TSCAN and determined that it creates no significant risk to the welfare of participants in this study or to the integrity of this research. B.A.S. is a member of the scientific advisory boards of Biotheryx and Proxygen. The remaining authors declare no competing interests.

## Additional information

[1]Department of Structural Biology, St. Jude Children's Research Hospital, Memphis, TN 38105, USA. [2]Department of Chemical Biology and Therapeutics, St. Jude Children's Research Hospital, Memphis, TN, USA. [3]Department of Cell Biology, Harvard Medical School, Boston, MA, USA. [4]Division of Genetics, Brigham and Women's Hospital, Howard Hughes Medical Institute, Department of Genetics, Harvard Medical School, Boston, MA, USA. [5]Department of Molecular Machines and Signaling, Max Planck Institute of Biochemistry, Martinsried, Germany. [6]These authors contributed equally: Daniel C. Scott, Suresh Dharuman. ✉e-mail: Richard.Lee@Stjude.org; schulman@biochem.mpg.de

