## [Peer Review File · Nature Communications]

Reviewers' Comments:

Reviewer #1:

Remarks to the Author:

Scott et al. developed new compounds for CRL2-KLHDC2 E3 ligase, with their moiety as the mimic of the di-Gly C-degron. They also uncovered the mechanism by solving several crystal structures of ligand-bound KLHDC2. Moreover, they designed potent PROTACs via linking their ligands with JQ1, a known ligand of BRD4. The PROTAC molecule could overcome the auto-inhibited KLHDC2 E3 ligase and efficiently induce the degradation of neo-substrate. Overall, this is a well-performed study that combines chemical biology work with structural biology, cell biology and biochemical data. The conclusion is convincing. I only have several minor concerns for them to improve the manuscript.

1. In all tables for crystallography data collection and refinement statistics, including Supplementary Table 2, 4, and 5. The α and γ for P21 space group should be indicated as 90 degree rather than 90.0. Also the redundancy values for both overall and high resolution shell should be provided, as shown for R-merge and CC1/2 values.
2. In Fig. 3b and 3e, the units for Kds and IC50 are not correctly shown.
3. In Fig. 5b, bine-a-fide should be "bona fide".
4. Figure legend for Supp. Fig. 3g, the level of Fo-Fc density should be 2.5 sigma or 2.5 σ .
5. Both KD and Kd are used to indicate the dissociation constant in the manuscript. A unified format is preferred.

Reviewer #2:

Remarks to the Author:

Review of Scott, Dharuman, et al.

This paper by Scott, Dharuman et al. presents a valuable study on the development of a novel series of small molecule recruiters for the E3 ligase CRL2-KLHDC2 and utilisation of this to develop cooperative degraders of BET bromodomain proteins with selectivity towards BRD3. This study complements other recent studies that establish KLHDC2 as an additional E3 substrate receptor amenable to be co-opted for Targeted Protein Degradation (TPD), an important advance for the field to further expand the E3 ligase repertoire for TPD. This work remains distinct and significant in validating a chemically differentiated KLHDC2 recruiting ligands with unique exit vectors and demonstrating using extensive biophysical, biochemical and cellular profiling to demonstrate that cooperative PROTACs recruiting KLHDC2 to BRD3 can be achieved. An ester pro-drug strategy is successfully utilised to address cell permeability limitations inherent to the

carboxylic acid pharmacophore of the binder, enabling relatively potent cellular BRD3 degradation that echoes selectivity in recruitment. Overcoming KLHDC2 auto-inhibition for neo-substrate recruitment is also examined, as is, in part, selectivity with respect to recruiting other substrate receptors that may overlap in C-end degron recognition (KLHDC3, KLHDC10, KLHDC1). Significantly, establishing cooperative target protein recruitment as a feasible optimisation strategy for degrader activity and selectivity via KLHDC2 has important implications for future KLHDC2-recruiting PROTAC development towards diverse disease-causing proteins beyond the BET family.

I congratulate the authors on a compelling study. The paper is well developed with a logical progression, clearly written and with generally very strong supporting data. Multiple techniques are utilised to characterise the binders/PROTACs, including SPR, TR-FRET, thermal shift, in vitro ubiquitination assays, X-ray crystallography, in addition to cellular degradation and global proteomics. In sum, this paper represents a significant contribution to the field of protein degradation and is well-suited for Nature Communications, albeit with some important comments noted below that in my view would be important to address for publication.

Major comments

1. The observation of relative selectivity by SJ46421 for cooperative recruitment of Brd3 BD2 to KLHDC2, relative to either Brd2 BD2 or Brd4 BD2, mediated by a single amino acid difference, is quite a striking one - particularly as noted as an interesting counterpoint to the opposite effect of the same residue on Brd3 selectivity previously observed for the VHL-based degrader MZ1 (Roy et al. 2019 ACS Chem Biol; Gadd et al. 2017 Nat Chem Biol). This finding implies this particular Glu residue of Brd3 BD2 is likely involved in a key interface interaction, potentially at one of the basic patches on KLHDC2 adjacent to the recruitment site (eg. Fig 3c). As the authors foreshadow, future structural studies would likely be informative to address this. The differences with respect to cooperative recruitment to KLHDC2 on the side of the different BD2 bromodomains are neatly addressed through ITC binding experiments; however, exploring this also on the KLHDC2 side seems perhaps a missed opportunity. Given the short PROTAC linker, known exit vector and well characterised structures of Brd3 BD2 with JQ1, perhaps it may already be narrowed down to a small number of residues on KLHDC2 most likely to mediate this (particularly with modelling) that can be evaluated using established assays either biophysically (eg. ITC/SPR/TR-FRET) or biochemically (competitive ubiquitination assay as for SJ46421 vs SJ46423). Perhaps the authors can comment on this.

2. Selectivity of E3 recruitment. Lines 259-262 and lines 341-342. Selectivity with respect to KLHDC2 over KLHDC3 or KLHDC10 or KLHDC1 was only evaluated with

respect to BRD3 BD2 ubiquitination. Perhaps the authors can comment whether direct or competitive binding towards purified KLHDC3/10/1 were measured for the KLHDC2 binders or PROTACs (eg. SPR or via thermal shift assay or alternatively competition for substrate recruitment or ubiquitination in the absence of a neo-substrate). As these are new KLHDC2 recruiting molecules, aside from the context of BRD3 ubiquitination specifically, it would be relevant for the field to benchmark the selectivity these molecules possess on the direct binding level.

3. The authors should ensure to include all key source data and all chemical characterisation data in the supporting information and source data file (eg. original ¹H and ¹³C spectra, for key final compounds in particular; all relevant fitting data for ITC and SPR should be included for all repeats, eg. tabulated R_{max} and fitted parameters for SPR, measured thermodynamic parameters and stoichiometry for ITC; uncropped blots etc.). For example: Supplementary Figure 3C-F tabulated fitted parameters for SPR/ITC are not shown; Figure line 223 Figure 4, ITC plots in Supplementary Figure 4 should be referred to in the main text, and individual fitted tabulated data summarised in source data.

4. Uncertainty/error estimates and exact number of independent repeats (n) should be specified for all data (eg. for thermal shift, SPR, TR-FRET, ITC). In many locations significant figures (SF) in KD/IC₅₀ values and some values in crystallographic tables are given to an unnecessarily high level of precision – suggest adjusting these to most appropriate SF based on uncertainty estimates. For example: Supplementary Table 1 error values are not stipulated; Line 325, Figure 6, independent repeats/errors are not stipulated.

5. Proofing of chemical structures and references to schemes. Line 188 to 190 (Fig 3d - SJ46421 and SJ46423) the figure shows the same chemical structure for each compound, where the exit vector should differ. Suggest to double check the annotation of these compounds throughout, as well as references to compounds in Schemes, eg. reference to Scheme 9 (line 190) appears to currently be incorrect. Naming of schemes in SI should also be checked (eg. Scheme S10 vs Scheme 9).

Minor comments:

BET bromodomain family members (BRD2/3/4) – a few instances should perhaps refer to BET paralogues rather than isoforms (title, abstract, Lines 11, 211, 216, 243, 244, 249)

Proofing note on Figures - in a few places in Figures the ‘micro’ symbol was lost.

File extension, “.pdb” in the PDB accession code is perhaps unnecessary to include.

In some instances in the text, PROTAC target selectivity and cooperativity are referred to together; while they may be linked in certain cases, they need not always be - for example, a PROTAC may not be positively cooperative yet may exhibit enhanced target selectivity relative to the constituent target binder (eg. via incompatible ternary complexes and/or lysine accessibility). Eg. Line 3-5, line 53, line 93, line 240 - perhaps the word selective is therefore not needed. Lines 55 to 57 - it might be worth noting that selective ternary complexes need not be cooperative and suggest to include some references regarding selectivity of cooperative and/or noncooperative ternary complexes.

Line 53-55 some recent references might be added regarding structural analysis of lysine accessibility.

Line 30 ‘catastrophic failure’ is rather a strong wording – perhaps consider rephrasing

Line 119. Check that SPR binding curves and fitted source data have been included. Suggest to include KD value in text. Line 132 to 134 suggest to include IC50 or KD values or temperature shift in brackets in the text.

Line 162 -163. Suggest to change/remove the last sentence speculating on the water mediated interaction as there’s no direct data/modelling provided regarding conformational flexibility of the ligand and the effect on binding profile.

Line 171-172. Suggest to check this sentence, as the wording is not clear and a word may be missing.

Line 191-193. Suggest for the reader to specify and reference the KLHDC2 modification not subject to auto inhibition.

Line 212-213. This sentence suggests that cooperative PROTACs are always more effective than non-cooperative PROTACs which necessarily the case; albeit that enhancing cooperativity in many cases can be an effective strategy to yield potent PROTACs. Perhaps consider rephrasing. Additional references could be included here and in Line 347, such as Farnaby et al. 2019 Nat Chem Biol.

Line 219. bracket missing

Line 221. Reference for cooperativity is needed.

Line 229-230. as ternary complex *affinity* with SJ 46423 is not measured directly perhaps they should refer to the competition assay; for example, comparatively weaker competition with endogenous substrate.

Line 245-248. Consider standardising nomenclature for amino acids referred to.

Line 222. Statement citing little to no reduction in the levels of BRD2 or BRD4 seems overstated as BRD4 also seems to be degraded in Fig 6 and later proteomic data. Perhaps this statement can be revised. Similarly, in line 305 and 306 'marginal degradation' seems is relatively strong regarding Fig 6 - perhaps noting partial degradation would be better.

Reviewer #3:

Remarks to the Author:

The authors are reporting a new ligand for KLDHC2 which can be used for PROTAC design. Using a thermal shift screening method, an initial hit with a structure similar to the known diGly degron was identified. Hit expansion SAR led to a sub-micromolar binder utilized for PROTAC design with JQ1. Structural work to identify an exit vector yielded two different feasible routes. Ultimately, the best structure lead to very high cooperativity in ternary complex formation between KLDHC2 and BRD3. After extensive in vitro characterization, the authors found a prodrug approach with esterification of the terminal acid yielded a selective PROTAC degrader of BRD3.

Overall, I feel this is a very detailed and high quality study to support usability of KLHDC2 for TPD and the conclusions appear to be in line with previously reported structures. Expanding the toolkit for E3 ligands can certainly expand diversity in the TPD field. The considerable characterization of SJ46421 will likely make it a more attractive starting point for others.

Comments/questions:

- Several hits showed a much more significant increase in T_m compared to SJ6145 – were any of these followed up on? What is the hit rate from the initial screen and percent validation?
- There is a relatively small SAR investigation around SJ6145 shown. What was the goal for binding before moving forward? Were there any efforts to replace the diGly motif with a peptidomimetic to improve stability and/or cell permeability (beyond the prodrug approach)?
- I like the structure-based design to pursue an exit vector similar to known endogenous

substrates. However, there is some confusion in both the manuscript (lines 188-189) and Fig. 3d – the structures provided are identical and the manuscript makes it appear the SJ46421 is the PROTAC using position 7 of the naphthyl ring for the linker. Based on the conclusions, this is not the case. Please clarify which is which and address the structures in the figure accordingly.

- To the authors' best knowledge, has such a design strategy to mimic native substrate binding position been used for ligands against other E3s?
 - I would suggest measuring the cooperativity of SJ46423 to be able to directly support the hypothesis that this is the reason for less potent inhibition of native substrate ubiquitination compared to SJ46423 (Figure 4c).
 - Fig 5d has a typo in the pre-incubation time in the graph.
 - For Figure 6b – can you quantify the % degradation? Was this experiment done in replicates? If yes, please provide an average level of degradation across experiments.
 - Given the release of auto-inhibition for KLHDC2, is there a significant increase in KLHDC2 observed during PROTAC treatment? How does this affect endogenous substrates if there is more active KLHDC2?
 - Were degradation treatments done in a time course? Does this then affect selectivity?
 - From Fig 6e, BRD3 and BRD4 degradation does not look very different. Similar to the previous point, have you done a shorter time point to see if this affects degree of selectivity?
 - There is relatively little comparison between the acid and prodrug PROTACS. I would suggest running the degradation in non-denatured lysate to see if the difference in activity can solely be explained by the poor cell permeability of the acid. I would also recommend some sort of cellular engagement or permeability experiment to measure amounts of the prodrug vs. acid able to get into the cell.
- Suggested additional experiments prior to publication:
- Measurement of cooperativity for SJ46423.
 - Time course for degradation with SJ46420 – either western blot or proteomics to cover BRD2, BRD3 and BRD4.
 - Degradation treatment in cell lysate to compare SJ46420 and SJ46421 in order to support conclusion that prodrug is majorly assisting in cell permeability.
 - Probably outside of the scope of this publication – an additional example beyond JQ1 to demonstrate if this ligand can be used more broadly in PROTAC design.

Reviewer #4:

Remarks to the Author:

In the submitted manuscript by Scott and colleagues the synthesis of new molecules binding the E3 ligase substrate adaptor KLHDC2 as basis for PROTACs is described. The ligands were derived from large-scale chemical screens and structure-activity-based

optimization. A highly selective KLHDC2 E3 ligase binder was generated and linking to a BRD binder resulted in a BRD family homolog-selective potent degrader. The work is original, the study is technically well performed and the results are mostly convincing. New ligase binders are extremely important for the TPD field to extend the number of exploitable E3 ligases and subsequently targetable substrates and the work is thus of interest for a broad readership. Another very recently published study (Scott et al. Nat Chem Biol 2024) also described KLHDC2 binders and PROTACs, yet the compounds and their binding mode are different and the current work provides sufficient novelty. Overall, this is a very good manuscript.

Comments:

- Title: BRD 2,3,4 are gene/ protein family members and not isoforms
- Figure 3d: the structure of the key compounds SJ46421 und SJ46423 are identical both linked at position 8, this is probably a confusion.
- The number of evaluated cell lines for BRD degradation (figure 6 and S6) is only 4 with half of them being prostate cancer cells. To generalize the effectivity of the PROTACs more cell lines also from other cancer types should be evaluated with different concentrations. In addition, a time course of BRD degradation should be performed in at least two cell lines.
- BRD protein degradation efficacy and selectivity of the best PROTAC from this paper and the published (Scott et al. Nat Chem Biol 2024) one should be directly compared in the same experiment and cell line to grasp their potency.
- Toxicity studies of the PROTACs on BRD-dependent and -independent cell lines are missing.
- Although there is some preference on degradation of BRD3 over BRD4, and 2, the latter two actually are degraded in the western blot and proteomic experiments. Protein levels should be quantified from repeat experiments to better assess the degradation effect on all three BRD proteins. The statement of selectivity should also be tuned down a bit.

Reviewer #5:

Remarks to the Author:

General response to reviewers:

We are pleased by the enthusiastic responses from the Reviewers! It is gratifying to receive such supportive comments regarding our work, thank you! We are also grateful to the Reviewers for helpful suggestions, which we believe have improved the manuscript. We tried to address all suggestions, both by performing experiments and revising the text and figures.

Reviewer #1 (Remarks to the Author):

Scott et al. developed new compounds for CRL2-KLHDC2 E3 ligase, with their moiety as the mimic of the di-Gly C-degron. They also uncovered the mechanism by solving several crystal structures of ligand-bound KLHDC2. Moreover, they designed potent PROTACs via linking their ligands with JQ1, a known ligand of BRD4. The PROTAC molecule could overcome the auto-inhibited KLHDC2 E3 ligase and efficiently induce the degradation of neo-substrate. Overall, this is a well-performed study that combines chemical biology work with structural biology, cell biology and biochemical data. The conclusion is convincing. I only have several minor concerns for them to improve the manuscript.

We thank the Reviewer for such kind comments and enthusiasm for our work!

1. In all tables for crystallography data collection and refinement statistics, including Supplementary Table 2, 4, and 5. The α and γ for P21 space group should be indicated as 90 degree rather than 90.0. Also the redundancy values for both overall and high resolution shell should be provided, as shown for R-merge and CC1/2 values.
2. In Fig. 3b and 3e, the units for Kds and IC50 are not correctly shown.
3. In Fig. 5b, bine-a-fide should be "bona fide".
4. Figure legend for Supp. Fig. 3g, the level of Fo-Fc density should be 2.5 sigma or 2.5 σ .
5. Both KD and Kd are used to indicate the dissociation constant in the manuscript. A unified format is preferred.

We thank the Reviewer for such careful reading of the manuscript and bringing these points to our attention. We have made all the suggested corrections.

Reviewer #2 (Remarks to the Author):

Review of Scott, Dharuman, et al.

This paper by Scott, Dharuman et al. presents a valuable study on the development of a novel series of small molecule recruiters for the E3 ligase CRL2-KLHDC2 and utilisation of this to develop cooperative degraders of BET bromodomain proteins with selectivity towards BRD3. This study complements other recent studies that establish KLHDC2 as an additional E3 substrate receptor amenable to be co-opted for Targeted Protein Degradation (TPD), an important advance for the field to further expand the E3 ligase repertoire for TPD. This work remains distinct and significant in validating a chemically differentiated KLHDC2 recruiting ligands with unique exit vectors and demonstrating using extensive biophysical, biochemical and cellular profiling to demonstrate that cooperative PROTACs recruiting KLHDC2 to BRD3 can be achieved. An ester pro-drug strategy is successfully utilised to address cell permeability limitations inherent to the carboxylic acid pharmacophore of the binder, enabling relatively potent cellular BRD3 degradation that echoes selectivity in recruitment. Overcoming KLHDC2 auto-inhibition for neo-substrate recruitment is also examined, as is, in part, selectivity with respect to recruiting other substrate receptors that may overlap in C-end degron recognition (KLHDC3, KLHDC10, KLHDC1). Significantly, establishing cooperative target protein recruitment as a feasible optimisation strategy for degrader activity and selectivity via KLHDC2 has important implications for future KLHDC2-recruiting PROTAC development towards diverse disease-causing proteins beyond the BET family.

I congratulate the authors on a compelling study. The paper is well developed with a logical progression, clearly written and with generally very strong supporting data. Multiple techniques are utilised to characterise the binders/PROTACs, including SPR, TR-FRET, thermal shift, in vitro ubiquitination assays, X-ray crystallography, in addition to cellular degradation and global proteomics. In sum, this paper

represents a significant contribution to the field of protein degradation and is well-suited for Nature Communications, albeit with some important comments noted below that in my view would be important to address for publication.

We are delighted by the Reviewer's enthusiasm and support of our work.

Major comments

1. The observation of relative selectivity by SJ46421 for cooperative recruitment of Brd3 BD2 to KLHDC2, relative to either Brd2 BD2 or Brd4 BD2, mediated by a single amino acid difference, is quite a striking one - particularly as noted as an interesting counterpoint to the opposite effect of the same residue on Brd3 selectivity previously observed for the VHL-based degrader MZ1 (Roy et al. 2019 ACS Chem Biol; Gadd et al. 2017 Nat Chem Biol). This finding implies this particular Glu residue of Brd3 BD2 is likely involved in a key interface interaction, potentially at one of the basic patches on KLHDC2 adjacent to the recruitment site (eg. Fig 3c). As the authors foreshadow, future structural studies would likely be informative to address this. The differences with respect to cooperative recruitment to KLHDC2 on the side of the different BD2 bromodomains are neatly addressed through ITC binding experiments; however, exploring this also on the KLHDC2 side seems perhaps a missed opportunity. Given the short PROTAC linker, known exit vector and well characterised structures of Brd3 BD2 with JQ1, perhaps it may already be narrowed down to a small number of residues on KLHDC2 most likely to mediate this (particularly with modelling) that can be evaluated using established assays either biophysically (eg. ITC/SPR/TR-FRET) or biochemically (competitive ubiquitination assay as for SJ46421 vs SJ46423). Perhaps the authors can comment on this.

The Reviewer raises an excellent point. Our attempts at modeling did not converge on a single or informative structural model. Thus, we took an alternative approach to address the Reviewer request to find KLHDC2 residues mediating the SJ46421-dependent interaction with BRD3^{BD2}. Examining the electrostatic potential of the surface of our KLHDC2 structure revealed four basic patches centered around KLHDC2 residues Arg56, Arg112, Lys291, and His346/Arg347. We mutated these residues to alanine and performed purifications from 12L culture of each mutant to obtain enough protein for 1-2 ITC experiments. Given the extremely high protein requirements for these experiments, we first surveyed all the mutants by measuring affinity in forming a ternary complex with SJ46421-BRD3^{BD2}. Of these, only the R56A mutation substantially impacted the ternary complex affinity. We next scaled up the purification of R56A (60 liters of culture) to obtain sufficient protein to measure in duplicate the affinity of this mutant for ligand only, ternary complex formation with BRD3^{BD2}, and the G>E mutant of BRD4^{BD2} which cooperatively interacts with KLHDC2. Gratifyingly, the data show that upon R56A mutation, the α values for cooperativity fell to 2.0 and 1.6 respectively (as compared to α values of 16 and 12.7 for WT KLHDC2). Thus, we were able to identify a KLHDC2 residue (R56) driving cooperative ternary complex formation. Given its positive charge, we speculate this could complement E344 in BRD3^{BD2}. The new data are now included in Fig. 4.

2. Selectivity of E3 recruitment. Lines 259-262 and lines 341-342. Selectivity with respect to KLHDC2 over KLHDC3 or KLHDC10 or KLHDC1 was only evaluated with respect to BRD3 BD2 ubiquitination. Perhaps the authors can comment whether direct or competitive binding towards purified KLHDC3/10/1 were measured for the KLHDC2 binders or PROTACs (eg. SPR or via thermal shift assay or alternatively competition for substrate recruitment or ubiquitination in the absence of a neo-substrate). As these are new KLHDC2 recruiting molecules, aside from the context of BRD3 ubiquitination specifically, it would be relevant for the field to benchmark the selectivity these molecules possess on the direct binding level.

The Reviewer requests further characterizing the selectivity of KLHDC2 ligands across the KLHDC family. Due to idiosyncratic features of some family members (often the case across multiple members of multiprotein complex families), we have not been able to establish a streamlined assay quantifying binding across the entire KLHDC family. Thus, we benchmarked selectivity using our pulse-chase format ubiquitylation assay. We monitored the concentration dependence of SJ46411 and SJ46421 in inhibiting substrate ubiquitylation. The comparison was possible because we could use a common substrate, SELK

(in unpublished work, we observed that a SELK fragment is a robust substrate for all KLHDC family members).

The KLHDC2 ligand, SJ46411 does not show substantial effects on any KLHDC family member other than KLHDC2. The results indicate at least two orders-of-magnitude selectivity towards KLHDC2. Similar results were obtained for the heterobifunctional molecule SJ46421, with the notable exception that it slightly affected KLHDC10. Nonetheless, SJ46421 still showed ~40 fold greater selectivity toward KLHDC2 over KLHDC10. These results are now included in Supplementary Figure 5.

3. The authors should ensure to include all key source data and all chemical characterisation data in the supporting information and source data file (eg. original ¹H and ¹³C spectra, for key final compounds in particular; all relevant fitting data for ITC and SPR should be included for all repeats, eg. tabulated R_{max} and fitted parameters for SPR, measured thermodynamic parameters and stoichiometry for ITC; uncropped blots etc.). For example: Supplementary Figure 3C-F tabulated fitted parameters for SPR/ITC are not shown; Figure line 223 Figure 4, ITC plots in Supplementary Figure 4 should be referred to in the main text, and individual fitted tabulated data summarised in source data.

We have provided all the chemical characterization data in the supplementary information, including ¹H and ¹³C spectra for key final compounds SJ6145, JS10278, SJ46411, SJ46418, SJ46420, SJ46421, SJ46422, and SJ46423.

Per the reviewer's suggestion, we now include Supplementary Tables 6 and 7 which tabulate parameters from SPR and ITC experiments.

4. Uncertainty/error estimates and exact number of independent repeats (n) should be specified for all data (eg. for thermal shift, SPR, TR-FRET, ITC). In many locations significant figures (SF) in KD/IC₅₀ values and some values in crystallographic tables are given to an unnecessarily high level of precision – suggest adjusting these to most appropriate SF based on uncertainty estimates. For example: Supplementary Table 1 error values are not stipulated; Line 325, Figure 6, independent repeats/errors are not stipulated.

We have added error estimates, in addition to the number of independent repeats throughout the manuscript in addition to adjusting the crystallographic tables to more reasonable SFs.

Per the reviewer's suggestions, we have provided the standard error for SPR, thermal shift, and TR-FRET data for all the compounds.

5. Proofing of chemical structures and references to schemes. Line 188 to 190 (Fig 3d - SJ46421 and SJ46423) the figure shows the same chemical structure for each compound, where the exit vector should differ. Suggest to double check the annotation of these compounds throughout, as well as references to compounds in Schemes, eg. reference to Scheme 9 (line 190) appears to currently be incorrect. Naming of schemes in SI should also be checked (eg. Scheme S10 vs Scheme 9).

We apologize for the confusion, and we are grateful to the Reviewers for spotting this inadvertent errors. We have now fixed the figure appropriately to represent the correct molecule. We have also corrected all the supplementary scheme names and numbers.

Minor comments:

BET bromodomain family members (BRD2/3/4) – a few instances should perhaps refer to BET paralogues rather than isoforms (title, abstract, Lines 11, 211, 216, 243, 244, 249)

We thank this Reviewer and Reviewer #4 for pointing out this error on our part. We have changed the title, abstract, and text to refer to the BET family members as paralogs.

Proofing note on Figures - in a few places in Figures the 'micro' symbol was lost.

We regret that we missed this formatting error during the initial submission. We have now corrected the figures so that they symbol is maintained.

File extension, “.pdb” in the PDB accession code is perhaps unnecessary to include.

We have removed the .pdb extension from the accession codes.

In some instances in the text, PROTAC target selectivity and cooperativity are referred to together; while they may be linked in certain cases, they need not always be - for example, a PROTAC may not be positively cooperative yet may exhibit enhanced target selectivity relative to the constituent target binder (eg. via incompatible ternary complexes and/or lysine accessibility). Eg. Line 3-5, line 53, line 93, line 240 - perhaps the word selective is therefore not needed. Lines 55 to 57 - it might be worth noting that selective ternary complexes need not be cooperative and suggest to include some references regarding selectivity of cooperative and/or noncooperative ternary complexes.

We added references regarding selective cooperative and noncooperative ternary complexes to the Introduction (previously lines 53-57), and reworded the sentences referred to above at line 93 and line 240.

Line 53-55 some recent references might be added regarding structural analysis of lysine accessibility.

We have now included references on roles of lysine selection by E3 ligases.

Line 30 ‘catastrophic failure’ is rather a strong wording – perhaps consider rephrasing

We have rephrased this to read “As they are turned over multiple times, PROTACs possess catalytic-like activity in cells prompting failure to the targeted biological system, which can only be reversed by resynthesis of the POI”.

Line 119. Check that SPR binding curves and fitted source data have been included. Suggest to include KD value in text. Line 132 to 134 suggest to include IC50 or KD values or temperature shift in brackets in the text.

As suggested, we have now included all related binding values in the text.

Line 162 -163. Suggest to change/remove the last sentence speculating on the water mediated interaction as there’s no direct data/modelling provided regarding conformational flexibility of the ligand and the effect on binding profile.

We removed this sentence as suggested.

Line 171-172. Suggest to check this sentence, as the wording is not clear and a word may be missing.

We have edited this sentence to read “Thus, the resultant exit vector from the KLHDC2-binding ligand could substantially impact cooperativity and PROTAC effectiveness and selectivity for POI engagement”.

Line 191-193. Suggest for the reader to specify and reference the KLHDC2 modification not subject to auto inhibition.

We have added the relevant reference.

Line 212-213. This sentence suggests that cooperative PROTACs are always more effective than non-cooperative PROTACs which necessarily the case; albeit that enhancing cooperativity in many cases can be an effective strategy to yield potent PROTACs. Perhaps consider rephrasing. Additional references could be included here and in Line 347, such as Farnaby et al. 2019 Nat Chem Biol.

We added the requested reference and rephrased this sentence based on the comments and new experiments performed to address Reviewer 4.

Line 219. bracket missing

Thank you for spotting this! We fixed this typo.

Line 221. Reference for cooperativity is needed.

We have added the relevant reference.

Line 229-230. as ternary complex *affinity* with SJ46423 is not measured directly perhaps they should refer to the competition assay; for example, comparatively weaker competition with endogenous substrate.

We have reworded this section of the manuscript as we now include new data. Prior to our initial submission we attempted to measure the binding parameters via ITC for SJ46423 binding to KLHDC2 alone and its respective ternary complex with BRD3^{BD2}. However, when experiments were run at protein/molecule concentrations utilized for SJ46421 the heat responses for ligand alone were not sufficient to draw conclusions due to reduced heats of binding and affinity. To obtain data requested by Reviewer #3, we needed to perform a 120 liter prep of KLHDC2, which yielded enough protein to re-run these titrations in duplicate at 4X concentrations of components (600 μ M KLHDC2 and 40 μ M SJ46423). The results yielded K_d values of 737 nM for ligand alone and 251 nM for ternary complex formation, yielding an α value of 2.9. Thus, the ternary complex promoted by SJ46423 is indeed less cooperative as compared to SJ46421. However, the ITC runs produced N values of \sim 0.5, presumably because of solubility issues with the molecule at a final concentration of 40 μ M, which also resulted in less than ideal isotherm responses. We now include these data in the manuscript in Fig 4a, while referencing the limitations of the ITC experiment in the text. We also provide additional supporting evidence of this through the competition assays referred to above (Fig. 4b,c).

Line 245-248. Consider standardising nomenclature for amino acids referred to.

Per the Reviewer's suggestion we have standardized the amino acid nomenclature throughout the manuscript.

Line 222. Statement citing little to no reduction in the levels of BRD2 or BRD4 seems overstated as BRD4 also seems to be degraded in Fig 6 and later proteomic data. Perhaps this statement can be revised. Similarly, in line 305 and 306 'marginal degradation' seems is relatively strong regarding Fig 6 - perhaps noting partial degradation would be better.

We thank the Reviewer for this suggestion. We have revised the text to be less bold and refer to the degradation of BRD2 and BRD4 as "partial" as suggested by the Reviewer.

Reviewer #3 (Remarks to the Author):

The authors are reporting a new ligand for KLDHC2 which can be used for PROTAC design. Using a thermal shift screening method, an initial hit with a structure similar to the known diGly degron was identified. Hit expansion SAR led to a sub-micromolar binder utilized for PROTAC design with JQ1. Structural work to identify an exit vector yielded two different feasible routes. Ultimately, the best structure lead to very high cooperativity in ternary complex formation between KLDHC2 and BRD3. After extensive in vitro characterization, the authors found a prodrug approach with esterification of the terminal acid yielded a selective PROTAC degrader of BRD3.

Overall, I feel this is a very detailed and high quality study to support usability of KLHDC2 for TPD and the conclusions appear to be in line with previously reported structures. Expanding the toolkit for E3 ligands

can certainly expand diversity in the TPD field. The considerable characterization of SJ46421 will likely make it a more attractive starting point for others.

Comments/questions:

- Several hits showed a much more significant increase in T_m compared to SJ6145 – were any of these followed up on? What is the hit rate from the initial screen and percent validation?

We attempted to follow up on all hits in the primary TSA screen, first validating in dose-response and subsequently via bio-orthogonal assays. As is the usual nature of chemical screening, some did not repeat, others caused assay interference, and some did not cross-validate by SPR. Based on these criteria, shown in supplementary data section, only two chemical series were advanced, and SJ6145 was prioritized based on its good SPR profile. More clarification is provided in the methods portion of the manuscript.

“The 8892-compound lead-like one library was screened for its ability to stabilize KLHDC2 using a sypro orange-based thermal stability single-point assay. Three percent of these compounds were cherry-picked for further analysis. These compounds were tested in triplicate, and seven were selected based on reproducibility and data quality (0.07% of the original screen). When tested in dose-response format, three of these compounds were eliminated based on data quality and potency of the ligands. The four remaining compounds gave strong dose-response data in the sypro orange-based thermal stability assay. When tested on a sypro orange independent thermal stability assay using the 350/330nm absorbance over a temperature range, two further compounds were eliminated. These findings were further validated with TR-FRET EC₅₀ values of >200μM for the four eliminated compounds. SJ6145 (our lead compound for this study) and SJ8152 were retained for further analysis. While SJ8152 showed a higher change in thermal stability, the potency of this compound was lower. Unlike SJ6145, the attempted optimization of SJ8152 did not improve the potency of this ligand.”

- There is a relatively small SAR investigation around SJ6145 shown. What was the goal for binding before moving forward?

The goal for binding before moving forward was to enhance the target binding affinity of SJ6145 to reach sub-micromolar levels, as measured by our comprehensive panel of biophysical assays. This objective is crucial for the progression of our drug development process. In our SAR (Structure-Activity Relationship) investigation, we have seen a promising trend towards achieving this goal. For expediency, here we only show compounds that are informative to the SAR analysis and exclude many compounds that were generated and were inactive. In total 125 compounds were generated. Our goal was to find efficient ligands with minimal peptidic nature that were amenable to PROTAC development

-Were there any efforts to replace the diGly motif with a peptidomimetic to improve stability and/or cell permeability (beyond the prodrug approach)?

Our leads replace the second glycine motif with a thiazole ring but maintain the primary glycine found in the SelK peptide. Efforts to replace the primary Gly motif with bioisosteric replacements failed, leading us to focus on the prodrug strategy. We rationalize this result by saying that KLHDC2 has evolved to recognize the diGly motif with high specificity due to its global role in regulating the proteome. Thus, the need to retain the primary Gly motif is not surprising.

- I like the structure-based design to pursue an exit vector similar to known endogenous substrates. However, there is some confusion in both the manuscript (lines 188-189) and Fig. 3d – the structures provided are identical and the manuscript makes it appear the SJ46421 is the PROTAC using position 7 of the naphthyl ring for the linker. Based on the conclusions, this is not the case. Please clarify which is which and address the structures in the figure accordingly.

We apologize for the confusion, and we are grateful to the Reviewers for spotting this inadvertent error. We have now fixed the figure appropriately to represent the correct molecule.

- To the authors' best knowledge, has such a design strategy to mimic native substrate binding position been used for ligands against other E3s?

The Reviewer asks a great question, but it's a difficult one to directly answer. There have been some studies investigating the exit vectors from VHL based PROTACS. These reports reveal that exit vectors do indeed play a role in promoting efficient target degradation. However, unlike KLHDC2 which binds substrates extreme C-termini connected by a flexible linker to a preceding globular domain, VHL binds hydroxyproline degrons in a more "flat" manner. Thus, in the absence of full-length substrate-VHL structures, and Lys mapping of targeted residues for ubiquitylation, it is difficult to ascertain if the best exit vectors of VHL PROTACS mimic the placement of natural Lys residues or not.

- I would suggest measuring the cooperativity of SJ46423 to be able to directly support the hypothesis that this is the reason for less potent inhibition of native substrate ubiquitination compared to SJ46423 (Figure 4c).

Prior to our initial submission we attempted to measure the binding parameters via ITC for SJ46423 binding to KLHDC2 alone and its respective ternary complex with BRD3^{BD2}. However, when experiments were run at protein/molecule concentrations utilized for SJ46421 the heat responses for ligand alone were not sufficient to draw conclusions due to reduced heats of binding and affinity. To obtain data for the Reviewer, we needed to perform a 120 liter prep of KLHDC2, which yielded enough protein to re-run these titrations in duplicate at 4X concentrations of components (600 μ M KLHDC2 and 40 μ M SJ46423). The results yielded K_d values of 737 nM for ligand alone and 251 nM for ternary complex formation, yielding an α value of 2.9. Thus, the ternary complex promoted by SJ46423 is indeed less cooperative as compared to SJ46421. However, the ITC runs produced N values of \sim 0.5, presumably because of solubility issues with the at a final concentration of 40 μ M, which also resulted in less than ideal isotherm responses. We now include these data in the manuscript in Fig 4a, while referencing the limitations of the ITC experiment in the text. We also provide additional supporting evidence of this through biochemical pulse-chase ubiquitylation assays (Fig. 4b,c).

- Fig 5d has a typo in the pre-incubation time in the graph.

We have corrected this typo in the graph.

- For Figure 6b – can you quantify the % degradation? Was this experiment done in replicates? If yes, please provide an average level of degradation across experiments.

We now provide a supplementary table that reports the averaged normalized levels +/- SD of BRD2, BRD3, BRD4, and KLHDC2 for all dosing experiments presented in the manuscript. The table is laid out to refer to each individual figure panel clearly stating the number of repetitions (n) from which the values were derived.

- Given the release of auto-inhibition for KLHDC2, is there a significant increase in KLHDC2 observed during PROTAC treatment? How does this affect endogenous substrates if there is more active KLHDC2?

The levels of endogenous KLHDC2 in all of the cell lines used increases only to a minimal degree, although we do observe greater increases in the levels of KLHDC2 in our U2OS knockout/rescue line. Although we have tested some hypotheses, none held up so we still do not know the underlying mechanisms. This could simply reflect that the levels of endogenously expressed KLHDC2 are so low that there is little free KLHDC2 at baseline undergoing auto-ubiquitylation and degradation. We were able to perform the requested experiment showing effects on the endogenous substrate USP1 (N-terminal fragment). This is affected by degrader treatment in our KO/rescue cell line (see figure panel below for the Reviewer). However, we are not comfortable drawing conclusions about this as we do not fully understand the pathways regulating KLHDC2 levels in cells.

- Were degradation treatments done in a time course? Does this then affect selectivity?
- From Fig 6e, BRD3 and BRD4 degradation does not look very different. Similar to the previous point, have you done a shorter time point to see if this affects degree of selectivity?

As noted above, we now include a supplementary table providing the average normalized values of proteins for all dosing experiments. In response to the Reviewer's question, and a similar question from Reviewer #4, we now provide time-course experiments in two different cell lines, U2OS and SKBR3. In SKBR3 cells the selectivity of BRD3 over BRD4 degradation goes from ~5 fold to ~1.5 fold from 8 hr to 24 hr. Similarly, selectivity peaks for BRD3 over BRD4 at 8 hr in U2OS at ~2.5 fold and falls to ~1.4 fold at 24 hr. The time-course data is now included in Supplementary Figure 6.

- There is relatively little comparison between the acid and prodrug PROTACS. I would suggest running the degradation in non-denatured lysate to see if the difference in activity can solely be explained by the poor cell permeability of the acid. I would also recommend some sort of cellular engagement or permeability experiment to measure amounts of the prodrug vs. acid able to get into the cell.

We performed the experiment requested by the Reviewer (see panel below). Non-denaturing lysates were prepared from our U2OS KO/rescue strain. Lysates were aliquoted to 4mg/ml final concentration and supplemented with DMSO, 10 μM SJ46420, or 10 μM SJ46421. Samples were incubated at 30° C and quenched at the indicated time points in SDS-PAGE sample buffer and processed for western blotting to monitor changes in the levels of BRDs. We did not observe a degrader dependent decrease in the levels of any of the BRD proteins. Interestingly, we did notice a relative instability of BRD3 that was largely reversed by the free acid, but not the prodrug. In addition, the free-acid seemed to prevent the insolubility of KLHDC2 in the lysates, as its levels significantly increased during the incubation period.

We would further note that permeability limitations for free acids across biological membranes are well established. Furthermore, substrate binding by KLHDC2 is strictly dependent on the substrate retaining a free C-terminus. Consistent with this, the prodrug SJ46420 is unable to promote ubiquitylation of BRD3^{BD2} in reconstituted biochemical assays.

Finally, we performed LC-MS-MS experiments to measure the concentration of SJ46421 in SKBR3 cells at various times after a 10 μ M dose of SJ46420 or SJ46421. The data demonstrate that, as expected, the free acid SJ46421, has poor cellular permeability, reaching a max concentration of ~11nM after a 24 hour exposure. On the other hand, SJ46420 is rapidly internalized with a cellular concentration of ~7200 nM after 4 hour exposure. However, this experiment revealed a relative instability of SJ46421 in cells from degradation and/or metabolism over time. The data are not included in Supplementary Table 9.

PROTAC	Dose time (hr)	Concentration (nM)
SJ46420	4	7227 +/- 477
SJ46420	8	371 +/- 25
SJ46420	24	26.5 +/- 4.7
SJ46421	4	5.6 +/- 0.46
SJ46421	8	7.9 +/- 0.39
SJ46421	24	11.3 +/- 0.77

Suggested additional experiments prior to publication:

- Measurement of cooperativity for SJ46423.
- Time course for degradation with SJ46420 – either western blot or proteomics to cover BRD2, BRD3 and BRD4.
- Degradation treatment in cell lysate to compare SJ46420 and SJ46421 in order to support conclusion that prodrug is majorly assisting in cell permeability.
- Probably outside of the scope of this publication – an additional example beyond JQ1 to demonstrate if this ligand can be used more broadly in PROTAC design.

We performed all the required experiments (a-c) on this list and the others requested above. While use of the ligand for additional degrader designs is an exciting suggestion for future work, we appreciate that the Reviewer recognizes that this is beyond the scope of the present study.

Reviewer #4 (Remarks to the Author):

In the submitted manuscript by Scott and colleagues the synthesis of new molecules binding the E3 ligase substrate adaptor KLHDC2 as basis for PROTACs is described. The ligands were derived from large-scale chemical screens and structure-activity-based optimization. A highly selective KLHDC2 E3 ligase binder was generated and linking to a BRD binder resulted in a BRD family homolog-selective potent degrader. The work is original, the study is technically well performed and the results are mostly convincing. New ligase binders are extremely important for the TPD field to extend the number of exploitable E3 ligases and subsequently targetable substrates and the work is thus of interest for a broad readership. Another very recently published study (Scott et al. Nat Chem Biol 2024) also described KLHDC2 binders and PROTACs, yet the compounds and their binding mode are different and the current work provides sufficient novelty. Overall, this is a very good manuscript.

We thank the Reviewer for their kind comments and enthusiasm for our work!

Comments:

-Title: BRD 2,3,4 are gene/ protein family members and not isoforms

As detailed above to Reviewer #2, we now correctly refer to BRD family members as paralogs throughout the manuscript.

-Figure 3d: the structure of the key compounds SJ46421 und SJ46423 are identical both linked at position 8, this is probably a confusion.

We apologize for the confusion, and we are grateful to the Reviewers for spotting this inadvertent error. We have now fixed the figure appropriately to represent the correct molecule.

-The number of evaluated cell lines for BRD degradation (figure 6 and S6) is only 4 with half of them being prostate cancer cells. To generalize the effectivity of the PROTACs more cell lines also from other cancer types should be evaluated with different concentrations. In addition, a time course of BRD degradation should be performed in at least two cell lines.

To address the Reviewer, we performed dosing data for 2 additional cell lines AU565 (epithelial cell line derived from a pleural effusion) and SKBR3 (breast cancer cell line). This brings the total number of cell lines for which our degraders were tested against in this study to 7.

-BRD protein degradation efficacy and selectivity of the best PROTAC from this paper and the published (Scott et al. Nat Chem Biol 2024) [sic Hickey et al. Nat Struct Mol Biol 2024] one should be directly compared in the same experiment and cell line to grasp their potency.

To address this, it was necessary to use the prodrug versions of two different PROTAC molecules, kindly provided by Arvinas: K2-B4-3e which corresponds to the star of the biochemical studies and the best biochemically-characterized of their KLHDC2-coopting PROTACs, and K2-B4-5e that showed superior efficacy in cells but its biochemical properties were not described. Comparing these alongside SJ46420 in two cell lines, U2OS and SKBR3, showed SJ46420 to be ~3 fold more efficacious than K2-B4-3e, while K2-B4-5e is ~4-fold better than SJ46420. Overall, these data highlight the complementarity of our study and that from Arvinas. These results are now included in Supplementary Figure 6.

-Toxicity studies of the PROTACs on BRD-dependent and -independent cell lines are still missing.

We chose five cell lines for toxicity studies based on their variable BRD family member dependencies (see below) as determined by depmap.org. (SKBR3-breast cancer line; AU565; epithelial cell line derived from a pleural effusion; RD-Rhabdomyosarcoma; SKES1-Ewings sarcoma; A673- Ewings sarcoma) Toxicity studies were performed via Cell Titer-Glo Luminescent Cell Viability assay (Promega; G7573) following a titration series of JQ1 or SJ46420. We observed similar IC₅₀ values for both JQ1 and SJ46420 (~300 nM) after 24 hours. Notably, the max response was typically much higher for SJ46420. Perhaps indicative of faster killing by SJ46420. However, given our discovery during revision of the limited stability of SJ46420 in cells after 24 hours, we cannot draw conclusions from these data, which are shown below for Reviewers.

DepMap dependency values for BRD family members in tested cell lines

	SKBR3	AU565	RD	SKES1	A673
BRD2	0.05	-0.56	-0.16	-0.17	-0.35
BRD3	-0.01	0.11	-0.07	-0.41	-0.04
BRD4	-1.26	-1.11	-0.86	-0.78	-1.33

	SKBR3	AU565	RD	SKES1	A673
JQ1 IC ₅₀ (uM)	0.53	0.30	1.53	0.87	0.41
JQ1 max effect	51%	75%	62%	53%	53%
SJ46420 IC ₅₀ (uM)	1.45	0.50	2.39	1.77	0.63
SJ46420 max effect	86%	91%	100%	100%	100%

-Although there is some preference on degradation of BRD3 over BRD4, and 2, the latter two actually are degraded in the western blot and proteomic experiments. Protein levels should be quantified from repeat experiments to better assess the degradation effect on all three BRD proteins. The statement of selectivity should also be tuned down a bit.

As noted above, we now include a supplementary table providing the average normalized values of proteins for all dosing experiments.

Reviewer #5 (Remarks to the Author):

We thank the Reviewer for the time and effort to review our manuscript.

Reviewers' Comments:

Reviewer #2:

Remarks to the Author:

I would again like to congratulate the authors on a highly interesting and thorough study of value to the field. They have done a commendable job to satisfactorily address the items I raised upon my initial review and those of other Reviewers - based on this I would not hesitate to recommend acceptance of the revised manuscript.

Reviewer #3:

Remarks to the Author:

I appreciate the amount of extra work and time put into this study based on my comments and suggestions and those of the other reviewers.

I don't have any additional comments or requests based on this new data. It was great to see the comparison to other KLHDC2 PROTACs and the work towards better understanding the effects of the ligand's effect on autot inhibition of KLHDC2. There is still a typo in Figure 5D - the second data point from the right is noted as "tetramer" but from the corresponding blots in Figure 5C, this should be "monomer."

Reviewer #4:

Remarks to the Author:

All my concerns/ questions were addressed by the authors and I have no further comments. I recommend acceptance for this highly important study.

Reviewer #5:

Remarks to the Author:

Response to Reviewers

We would like to thank all the reviewers for their positive comments regarding our manuscript NCOMMS-24-22496A, entitled "Principles of paralog-specific targeted protein degradation engaging the C-degron E3 KLHDC2." We are very pleased that all five reviewers recommend publication in Nature Communications.